



# Regional validation of the solar irradiance tool SolaRes in clear-sky conditions, with a focus on the aerosol module

Thierry Elias[(1)(*)], Nicolas Ferlay[(2)], Gabriel Chesnoiu[(2)], Isabelle Chiapello[(2)], Mustapha Moulana[(1)]

[(1)] HYGEOS, Euratechnologies, 165 boulevard de Bretagne, 59000 Lille, France

[(2)] Laboratoire d'Optique Atmosphérique, Université Lille, CNRS, UMR 8518, 59000 Lille, France

[(*)] Corresponding author: te@hygeos.com

**Abstract**

The objective of the paper is to validate SolaRes (Solar Resource estimate) in clear-sky conditions, and to examine the aerosol influence on the differences between observation and estimate. SolaRes has the ambition to fulfil both research and industrial applications exploiting downwelling solar
radiation at surface level. Consistently with solar resource applications, we show the capacity of SolaRes to reproduce the angular behaviour of the angular field, by validating not only global horizontal irradiance (GHI), but also direct normal irradiance (DNI), diffuse horizontal irradiance (DifHI), global and diffuse irradiance in tilted plane (GTI, DifTI), and even the circumsolar contributions.

Computations are made with the SMART-G radiative transfer code, taking spectral aerosol optical thickness (AOT) data sets as input, which are delivered by the Aerosol Robotic network (AERONET) and the Copernicus Atmospheric Monitoring Service (CAMS). A mixture of two aerosol models is required to compute aerosol optical properties. Measurements for validation are made at two sites in Northern France. Clear-sky is identified by two methods to show its influence:
1) a method reproducing the *AOT* variability conditions, and 2) a stricter method eliminating some residual cloud influence but also conditions with largest *AOT*.

SolaRes is validated according to comparison scores found in the literature, with the (relative) root mean square difference (RMSD) in *GHI* as low as 1%, and the mean bias difference (MBD) which could be 0%. Angular behaviour is reproduced with satisfying scores. The circumsolar contribution
improves *MBD* in *DNI* and *DifHI*, by 1% and 4% respectively, as well as *RMSD* by ~0.5%. *MBD* in *DNI* is around -1% and *RMSD* around 2%, and *MBD* in *DifHI* is 2% and *RMSD* around 9%. *RMSD* and *MBD* in both *DNI* and *DifHI* are larger than in GHI because they are more sensitive to the aerosol and surface properties. *DifTI* measured in a vertical plane facing South is reproduced with a *RMSD* of 8%, similar to *DifHI*. It is suggested a strong influence of reflection by not only ground
surface but also surrounding buildings, increasing the albedo from 0.13 to 0.35.

The sensitivity study on the aerosol parameterisation shows that the spectral *AOT* contains enough information for best quality in *DNI* retrieval. The choice of aerosol models in the parameterisation has an influence in *RMSD* smaller than 0.7%. Complementary information on angular scattering and aerosol absorption has a significant influence in *GHI* by reducing *RMSD* by ~0.5%, and *MBD*
by ~0.8%. The uncertainty on the data source has a significant influence. The CAMS data set increases the *RMSD* in *DNI* by 5%, but has has less influence in *GHI*, by increasing *RMSD* by ~1% and *MBD* by ~0.4%. *RMSD* in *GHI* still remains slightly smaller than state-of-the-art methods.



**footnotes**:

(1) https://www-loa.univ-lille1.fr/observations/plateformes.html?p=lille

(2) *https://ads.atmosphere.copernicus.eu/cdsapp#!/dataset/cams-global-atmospheric-composition-forecasts?tab=form*

(3) *https://www.soda-pro.com/web-services/radiation/cams-mcclear*




## 1. Introduction

Solar radiation incident on the collecting system is one of the main drivers of the electrical production by a solar plant. Incident solar radiation is highly variable because of changing atmospheric optical properties affected by clouds, aerosols, water vapour, ozone, and because of surface reflection. The electricity production also depends on the panel orientation and inclination, and on the spectral absorption efficiency.

We conceived the Solar Resource estimate tool (SolaRes) to provide precise and accurate simulations of the solar resource components for any location on the globe, in any meteorological and ground surface conditions, for any solar plant technology, and at the finest time resolution. SolaRes consequently suits many applications from research to industrial fields. SolaRes is powered by the Speed-up Monte Carlo Atmospheric Radiative Transfer code using GPU (SMART-G) which resolves physically the radiative transfer equation [Ramon *et al.*, 2019]. Until now, a physical radiative transfer code was rarely used to respond to industrial needs in solar energy [e.g. Sun *et al.*, 2019] because it is usually slower than approaches based on abaci or look-up tables. However SMART-G is fastened thanks to parallelisation on GPU cards, and advances in computing science make it a suitable tool. Such an approach offers the advantage of precision and accuracy, as well as flexibility, and radiative transfer can even be simulated in a complex physical environment embedded in a realistic changing atmosphere, even considering 3D interactions between solar radiation and the environment. Moulana *et al.* [2019] present preliminary work on the increased precision on solar resource in a tower concentrated thermal solar plant using SMART-G, and Moulana *et al.* [Submitted] present the technology to adapt SMART-G to consider reflection with 3D objects.

According to Lindsay *et al.* [2020], 15% error in simulated electrical power produced by PV could be avoided by computing spectrally-and-angularly refined irradiances, as can be done by SMART-G. SMART-G could be ranked in the class A of the solar resource model classification defined by Gueymard and Ruiz-Arias [2015], reviewing the performance of 24 radiative models from the literature. Indeed, any angular and spectral characteristics of the solar radiation field can be computed on demand by SMART-G. *DNI* and *DifHI* are computed separately to provide *GHI*, which can be both of importance in other fields such as vegetation processes. Also, the circumsolar contribution can be computed, providing two estimates of *DNI*: 1) $DNI_{pyr}$ consistent with observed *DNI*, including circumsolar contribution; 2) $DNI_{strict}$, not including circumsolar contribution, but which is consistent with computations of solar resource parameters in any panel orientation. Usually, physical or semi-physical models provide only one of these two estimates of *DNI*. For example Gueymard and Ruiz-Arias [2015] remind that circumsolar contribution is not considered by the 24 presented models.

This paper presents the validation of SolaRes in a 1D mode, providing not only the global horizontal irradiance (GHI) as the standard solar resource component, but also other components depending on the angular behaviour of the radiaton field, as direct normal irradiance (DNI) and the diffuse horizontal irradiance (DifHI), the circumsolar contributions, as well as the projected quantities on a tilted plane, i.e. the global tilted irradiance (GTI) and the diffuse tilted irradiance (DifTI). SolaRes encompasses the Attenuation of Solar Radiation by Aerosols (ASoRA) method for *DNI* estimates, which is validated in clear-sky conditions in an arid environment [Elias *et al.*, 2021].

As computation uncertainties come from both the model and the input data set, the validation must be performed with the input data set defined with the best precision. Aerosol optical thickness (AOT) can be measured with high precision thanks to the ground-based photometers contributing to the Aerosol Robotic NETwork (AERONET) [Holben *et al.*, 1998], evaluating the attenuation of the direct solar radiation in several narrow spectral ranges. However this is not the case for the clouds. Therefore, the validation is performed in clear-sky conditions, when aerosols affect the surface solar





irradiance but not the clouds. A major process thus consists in identifying the clear-sky moments in
highly variable overcast conditions. Many methods are presented in literature [e.g. Gueymard *et al.*,
2019]. We select and adapt two methods presenting contrasted results in terms of representativity of
the atmospheric variability and of comparison scores. The ambition of SolaRes is to reproduce the
impact of any atmospheric condition at the finest time resolution, which is 1 minute nowadays.
Consequently, we select a cloud-screening method missing a minimum number of clear-sky
moments and representing the full *AOT* variability, and another cloud-screening method avoiding
residual cloud influence but also missing some *AOT* variability. Whatever the method, more than 10
000 clear-sky moments could be selected per year.

The field of study of solar energy benefits of other research areas such as the climate studies. Some
of the measurements of solar radiation used as ground-based proof for validation are acquired by
the Baseline Surface Radiation Network (BSRN) [Driemel *et al.*, 2018], which had for first mission
to monitor components of the Earth's radiative budget, and their changes with time, with the
"*increasing debate on anthropogenic influences on climate processes during the 1980s*" [Driemel *et al.*, 2018]. In the same field, AERONET aims to evaluate the aerosol radiative forcing, partly
counteracting the greenhouse warming. This paper presents a radiative closure study as two
categories of independent measurements are related by a radiative transfer code [e.g. Michalsky *et al.*, 2006; Ruiz-Arias *et al.*, 2013]. The validation is performed on data sets acquired during two
years at two sites of northern France.

The main impact of aerosols is to attenuate the direct component of the solar radiation incident at
surface level. Input spectral AOT constrains efficiently this impact [Elias *et al.*, 2019; Elias *et al.*,
2021] as it depends on aerosol load and nature, aerosol nature driving the *AOT* spectral dependency.
However input spectral AOT poorly constrains the aerosol-scattering proportion which significantly
affects *DifHI*. A sensitivity study is performed which shows the impact of aerosol models
reproducing the input spectral *AOT*. The data source is also evaluated by changing to a global
product.

*Section 2* describes the observational and modelling data sets used as input of SolaRes, as well as
the solar irradiance measurements used as ground-based proof for validation. *Section 3* briefly
describes SMART-G, and the parameterisations used in SolaRes, especially that related to aerosol
contribution. *Section 4* investigates two cloud-screening procedures, and their impact on the
validation data base made by the solar resource parameters, and on the radiative factors such as
AOT and the water vapour content. *Section 5* presents the comparison scores obtained between
SolaRes estimates and solar irradiance ground-based measurements, for the validation of SolaRes.
Eventually, *Sect. 6* shows the sensitivity of the comparison scores on the aerosol parameterisation
considering two main influences: 1) the hypothesis on main aerosol nature, 2) the aerosol data
source. Indeed the Copernicus Atmospheric Monitoring Service (CAMS), assimilating satellite data
sets to describe air quality, is also used here as input data of SolaRes to show the sensitivity of
clear-sky estimates on the input data source.

## 2. Data

Our analysis of SolaRes performances relies on different types of data. Solar resource computations
require input data provided either by a ground-based instrumentation network (*Sect. 2.2*), either by
a global atmospheric model (*Sect. 2.3*). The SolaRes estimates are validated (*Sect. 5*) by making
comparisons between ground-based measured (*Sect. 2.1*) and computed solar resource components
(*Sect. 3*).





### 2.1. Ground-based irradiance measurements used as a validation data set

Two platforms located in northern part of France are chosen, both hosting a comprehensive set of radiative instruments.

*2.1.1. The ATOLL (ATmospheric Observations in LiLLe) platform*

Since 2008, a set of class A Kipp&Zonen instruments mounted on an EKO sun tracker (STR-22) measures routinely the solar downward irradiance at Villeneuve d'Ascq (France, 50.61167°N, 3.141670°E) on the ATOLL (ATmospheric Observations in LiLLe) platform, at the campus of Lille University[(footnote 1)]. A CHP1 pyrheliometer (Kipp & Zonen, 2008) measures the direct normal
irradiance ($DNI_{obs}$), in a field of view of 5±0.2°. A CMP22 pyranometer (Kipp & Zonen, 2013) associated with a shadowing ball measures the diffuse horizontal irradiance ($DifHI_{obs}$). Both *$DNI_{obs}$* and *$DifHI_{obs}$* are provided at 1-minute resolution.

Calibrations performed in 2012, 2017 and 2022 show a relative stability of the instrument performances. Indeed the CHP1 calibration coefficient varies by a maximum of 3% over the period,
and the CMP22 calibration coefficient decreases by less than 1%. According to Witthuhn *et al.* [2021], the uncertainty under clear-sky conditions is 2% for *GHI* and 4% for *DifHI*, considering uncertainty in shadowing device. Winter gaps exist in the data time series as the instruments of ATOLL are sent that season either in Delft (Netherland) for a recalibration (by Kipp and Zonen) or in M'Bour (Senegal) to be used as references for calibration of local instruments.

Observed global horizontal irradiance ($GHI_{obs}$) at Lille is obtained as the sum of direct and diffuse components, which is the preferred method for the measurement of global irradiance [Flowers and Maxwell, 1986], avoiding most cosine errors at low sun angles, and chosen by BSRN [Ohmura *et al.*, 1998]:

$$GHI_{obs} = DirHI_{obs} + DifHI_{obs}, \qquad (1a)$$

$$\text{with } DirHI_{obs} = DNI_{obs}\,\mu_0 \qquad (1b)$$

where $\mu_0 = \cos(SZA)$, and *SZA* is the solar zenith angle.

Additionally an unshaded class A Kipp&Zonen CMP11 pyranometer is in operation on ATOLL
since 2017 in variable inclinations, in order to measure the global tilted irradiance ($GTI_{obs}$). Both the CHP1 and CMP22 instruments measure radiation in the broadband range between 210 and 3600 nm, while the spectral range for the CMP11 pyranometer extends between 270 and 3000 nm.

Michalsky *et al.* [1999] show a possible range of 30 W/m$^2$ (> 5%) in *$GHI_{obs}$* between unshaded pyranometers because of cosine errors, and that uncertainty is multiplied by 2 to 3 with unshaded
pyranometers. The CMP11 is set horizontally during two 22-day and 49-day time periods in spring-summer 2018 for an intercomparison campaign with both CHP1 and CMP22. Comparison is made over 47 days with clear-sky moments according to the Garcia cloud-screening method (*Sect. 4*). The mean relative difference between *$GHI_{obs}$* measured by the CMP11 and by the CHP1+CMP22 instruments is 8±5 W/m$^2$ (1.6±0.9%) (CMP11 providing smaller values than CHP1+CMP22), and
the root mean square difference (RMSD) is 9 W/m$^2$ (1.9%).

The 2018-2019 time period is chosen for the paper, close to the 2017 calibration which shows instrument performance stability, including 2018 to benefit from the intercomparison campaign, and including 2019 to validate SolaRes in different angular configurations.



*2.1.2. BSRN site of Palaiseau*

Solar resource measurements are made at Palaiseau (France, 48.7116°N, 2.215°E) as part of BSRN, by three Kipp&Zonen CHP1 and CMP22 instruments, similar to those in Lille. $GHI_{obs}$ and $DNI_{obs}$ are measured by CMP22 and CHP1, respectively, and $DifHI_{obs}$ is measured by a second CMP22 mounted with a sun-tracking shadower device. A 1-Hz sampling rate is recommended for radiation
monitoring, and measurements are recorded and provided at 1-minute time resolution. Uncertainty requirements for the 1-min BSRN data are 5 W/m² for $DifHI_{obs}$, and 2 W/m² for $DNI_{obs}$ [Ohmura *et al.*, 1998].

### *2.2. Input data sets about aerosol and water vapour: AERONET*

AERONET provides the aerosol and water vapour input data processed by SolaRes in this paper. Indeed, at both sites, the AERONET photometers acquire measurements coincidentally with the pyranometers and pyrheliometers. In this study, we use direct measurements of aerosol optical thickness (AOT) at both 440 and 870 nm, as well as the column water vapour content (WVC) [Elias *et al.*, 2021]. The expected uncertainty in *AOT* is 0.01-0.02 at these wavelengths [Dubovik *et al.*,
2000; Giles *et al.*, 2019]. *AOT* measurements are made at the time resolution of around 3 minutes [Giles *et al.*, 2019], in clear-sun conditions. We perform 15-minute averages of these measurements in order to reduce the number of radiative transfer computations over a year (***Sect. 3***). We use the Level 2.0 data quality , and the V3 version of AERONET data [Sinyuk *et al.*, 2020], which also provides ozone content from "*Total Ozone Mapping Spectrometer (TOMS) monthly average*
*climatology (1978–2004)*".

AERONET provides not only measurements of *AOT* at several wavelengths but also inverted aerosol models at around 1 hour resolution. Given the high time variability of aerosols and of their influence on solar radiation, the time resolution is an important factor in solar resource estimation, and we choose for validation of SolaRes (***Sect. 5***) to rely on *AOT* acquired at around 3 minute
resolution. However we use the inverted aerosol model in ***Sect. 6*** to check the influence of the SolaRes aerosol parameterisation. As the Level 2.0 inverted data set is too sparse, we choose to use the Level 1.5 data quality [Ruiz-Arias *et al.*, 2013; Witthuhn *et al.*, 2021], with possible inconvenient on solar resource precision. Sinyuk *et al.* [2020] estimate an uncertainty of ±0.03 on the aerosol single scattering albedo (SSA), but according to Ruiz-Arias *et al.* [2013], the uncertainty
of Level 1.5 *SSA* increases to the 0.05–0.07 range. The option "hybrid" radiance products is chosen.

The averaged *SSA* at Lille in 2018 is 0.97±0.03 at 440 nm, 0.96±0.04 at 675 nm, and 0.95±0.04 at 870 nm, depicting little absorption.

### *2.3. Input data sets about aerosol, water vapour, and surface albedo: CAMS*

Data from the Copernicus Atmosphere Monitoring System (CAMS) [Benedetti *et al.*, 2009; Morcrette *et al.*, 2009] are used to investigate the sensitivity of SolaRes to the aerosol data source (***Sect. 6.3***). To be consistent with a near real time (NRT) service, the CAMS-NRT data set is used. *AOT* is provided at several wavelengths, as well as WVC. The spatial resolution is 0.4°, and the time resolution is 1 hour, considering the forecast mode between the two 12-hour runs. For the
paper, global CAMS-NRT data sets are downloaded from the Atmosphere data Store[footnote 2]. CAMS-NRT *AOT* at 469 and 865 nm are used to compute the Ångström exponent, that allows to infer *AOT* at both 440 and 870 nm (see for example Witthuhn *et al.* [2021]), as required by the SolaRes algorithm (***Sect. 3***). The comparison with AERONET direct measurements gives a *RMSD* of ~50% in *AOT* (0.10 at 440 nm, and 0.04 at 870 nm), and of 25% (0.3) for the Ångström
exponent. The *MBD* is smaller than 5% in *AOT* and for the Ångström exponent. These comparison





results are similar to Witthuhn *et al.* [2021] and references therein, but for Germany and the CAMS reanalysis data set.

CAMS-NRT data time series at Lille and Palaiseau are also downloaded from the CAMS-radiation service[(footnote 3)]. The 'research mode' allows to download not only *GHI*, *DNI*, and *DifHI*, but also the input data as *AOT, WVC*, the ozone content, as well as the surface albedo which is derived from MODIS as described by Lefèvre *et al.* [2013]. Surface albedo is taken from the CAMS-radiation service. Daily averages are computed, varying between 0.12 in November-December and 0.16 in June-July at Lille and Palaiseau.

**3. The SolaRes algorithm**

Computations are made with the SolaRes V1.5.0 algorithm. SolaRes computes *DNI* according to the ASoRA method [Elias *et al.*, 2021], and the diffuse irradiance with the SMART-G code [Ramon *et al.*, 2019], using a common input data set. The advantage in using SMART-G is to compute precisely the angular behaviour of the diffuse radiation field, by considering aerosol and surface optical properties: *DifHI* can be computed as well as *DifTI* for any inclination and orientation, and the circumsolar contribution can be estimated by computing the diffuse irradiance in a narrow field of view centred on the solar direction.

To better reproduce the solar resource time variability, and to better evaluate the performance of SolaRes in clear-sky conditions, computations are made at a 1-minute time resolution, as advised by several authors as Sun *et al.* [2019]. *DNI* is computed at the time resolution of 1 minute by interpolating the aerosol extinction properties at 1 minute. *DifHI* is computed at 15-minute by radiative transfer computations with SMART-G. It is then interpolated linearly at the 1-minute resolution. *GHI* is computed by adding 1-minute *DNI* projected on the horizontal plane (*DirHI*) and 1-minute *DifHI*, as done by all high-performance models referenced by Sun *et al.* [2019], and similarly for *GTI*:

$$GHI = DirHI + DifHI \tag{2a}$$
$$GTI = DirTI + DifTI \tag{2b}$$

Computations are made using AERONET spectral *AOT* (***Sect. 2.2***) for validation purposes (***Sect. 5 and 6***) and with CAMS-NRT spectral *AOT* (***Sect. 2.3***) for sensitivity study on the aerosol data source (***Sect. 6***).

***3.1. The direct contribution***

*3.1.1. DNI$_{strict}$, and its projection*

While *DifHI* and *DifTI* are computed with SMART-G (***Sect. 3.2***), *DirHI* and *DirTI* are computed by projecting *DNI* on a horizontal or tilted plane:

$$DirTI = DNI \ \vec{\Omega}_{s} \cdot \vec{n} \tag{3}$$

with $\vec{\Omega}_{s}$ the unit vector in the solar direction:





$$\vec{\Omega}_S = \left( \sin(SZA)\cos(SAA) ; \sin(SZA)\sin(SAA) ; \cos SZA \right) \quad, \tag{4}$$


where $SAA$ is the solar azimuthal angle. $\vec{n}$ is the unit vector perpendicular to the tilted surface:

$$\vec{n} = \left( \sin i \cos o ; \sin i \sin o ; \cos i \right) \quad, \tag{5}$$

where $i$ is the inclination of the tilted surface and $o$ its orientation. If the plane is horizontal, i=0, $\vec{\Omega}_S \cdot \vec{n} = \cos(SZA)$ , and DirHI = DNI $\mu_0$ (**Eq. (1b)**).

$DNI$ can either be $DNI_{strict}$ according to the 'strict' definition given by Blanc *et al.* [2014], either be $DNI_{pyr}$ as it is observed by a pyrheliometer. For $DNI_{strict}$, only beams in the solar direction are 290    counted, which are not scattered by the atmosphere. In other words, the circumsolar radiation is not accounted for. Underestimation of $DNI_{obs}$ by $DNI_{strict}$ is then expected. Consistently with the ASoRA method [Elias *et al.*, 2021], $DNI_{strict}$ is expressed as:

$$DNI_{strict} = F_{ESD} \int_{\lambda_{inf}}^{\lambda_o} E_{sun}(SZA, \lambda) \, T_{col}(SZA, \lambda) \, d\lambda \quad. \tag{6}$$


$F_{ESD}$ is the Earth-Sun distance correcting factor. The spectral integration is made between the two wavelengths $\lambda_{inf}$ and $\lambda_{sup}$. $E_{Sun}(\lambda)$ is the extra-terrestrial solar irradiance at the wavelength $\lambda$. $T_{col}(SZA, \lambda)$ is the atmospheric column transmittance, which can be decomposed as:

$$T_{col}(\lambda) = T_{Ray}(\lambda) \cdot T_{gas}(\lambda) \cdot T_{aer}(\lambda), \tag{7}$$

where $SZA$ is omitted for clarity. $T_{Ray}(\lambda)$ is the transmittance caused by Rayleigh scattering, along the atmospheric column, while $T_{gas}(\lambda)$ is caused by absorbing gases. Main variable absorbing gases in the atmospheric column are water vapour and ozone. In clear-sky conditions, $T_{col}(\lambda)$ does not 305    depend on the cloud transmittance. $T_{aer}(\lambda)$ is defined according to the Beer-Lambert-Bouguer law as:

$$T_{aer}(\lambda) = e^{-m_{air} \, AOT(\lambda)} \quad. \tag{8}$$

where $m_{air}$ is the optical air mass which can be approximated by $1/\mu_0$, and must take into account the Earth's sphericity for $SZA$ above 80° [e.g. Kasten and Young, 1989].






### 3.1.2. Considering the circumsolar contribution

The pyrheliometer measures not only beams in the solar direction but also all scattered radiation within the instrument field of view. The comparison scores are then expected to be improved by considering $DNI_{pyr}$ defined as:


$$DNI_{pyr} = DNI_{strict} + \Delta DifNI_{circ}, \tag{9}$$

where $\Delta DifNI_{circ}$ is the circumsolar contribution on a plane perpendicular to the solar direction. The sun-tracking shadowing device, allowing to measure $DifHI$ instead of $GHI$, does not block only direct radiation but also radiation scattered around the sun. $DifHI_{pyr}$ is then defined as:


$$DifHI_{pyr} = DifHI_{strict} - \Delta DifHI_{circ} \quad , \tag{10}$$

with


$$\Delta DifHI_{circ} = \Delta DifNI_{circ} \, \mu_0 \tag{11}$$

### 3.2. Brief description of SMART-G

SMART-G allows to simulate the propagation of polarised light (monochromatic or spectrally
integrated), in a coupled atmosphere-ocean system in a plane-parallel or spherical-shell geometry, as described by Ramon *et al.* [2019]. The code uses General-Purpose Computation on Graphic Processing Units technology with other Monte Carlo variance reduction methods (local estimation [Marchuk *et al.*, 1981], ALIS [Emde *et al.*, 2011], etc.) to speed up the simulations while keeping high precision.

In this work SMART-G is used to simulate all diffuse irradiance parameters i.e. *DifHI, DifTI*, and *ΔDifNI_{circ}*, in a plane-parallel atmosphere. *DifHI* is calculated by using the simple conventional method for planar flux in Monte Carlo radiative transfer codes, where the solar rays are tracked from the sun to the ground. The scattered rays reaching the ground surface are then counted to calculate *DifHI*. For *DifTI* we use a backward Monte Carlo tracking of solar radiation i.e. the solar
radiation rays are followed in the inverse path, from the instrument to the sun, with the local estimation method [Marchuk *et al.*, 1981] to reduce the variance. The half aperture angle is 90° to imitate the pyranometer. The circumsolar contribution *ΔDifNI_{circ}* is calculated similarly to *DifTI* but by assigning a half aperture angle of 2.5° to imitate the pyrheliometer.

### 3.3. The radiative transfer parameterisation

#### 3.3.1. Atmospheric gases and the surface

The extra-terrestrial solar spectrum is taken from Kurucz [1992]. Rayleigh optical thickness is computed according to Bodhaine *et al.* [1999], and scaled with the atmospheric pressure. The gas and thermodynamic profiles are adopted from the AFGL US summer standard atmosphere
[Anderson *et al.*, 1986], providing the water vapour optical thickness, which is scaled linearly with WVC from the input data source. Ozone and $NO_2$ absorption cross sections are taken from Bogumil *et al.* [2003], and we use the absorption band parameterisation provided by Kato *et al.* [1999] for





other gases like $H_2O$, $CO_2$, $CH_4$. As UV-C radiation below 280 nm is absorbed by the atmosphere, spectral integration is made for spectral bands between 280 and 4000 nm for comparisons with CHP1 and CMP22 measurements (297 g-points in Kato parameterisation), and between 280 and 3000 nm for comparisons with CMP11 measurements (267 g-points). Surface reflection is modelled by the surface albedo, considered spectrally independent.

### 3.3.2. Aerosol parameterisation

The spectral aerosol optical properties are computed at the wavelengths of the Kato parameterisation, according to Mie theory, as *AOT*, the aerosol phase function and single scattering albedo. Several aerosol models of the Optical Properties of Aerosols and Clouds (OPAC) database [Hess *et al.*, 1998] are used, as done in the ASoRA method [Elias *et al.*, 2021]. To compute *DNI*, two OPAC aerosol models AM1 and AM2 are mixed to reproduce the input *AOT* at two wavelengths, such as:

$$AOT_{input}(\lambda_1) = w_{AM1} \, AOT_{AM1}(\lambda_1) + w_{AM2} \, AOT_{AM2}(\lambda_1) \tag{12a}$$

$$AOT_{input}(\lambda_2) = w_{AM1} \, AOT_{AM1}(\lambda_2) + w_{AM2} \, AOT_{AM2}(\lambda_2) \tag{12b}$$

where *$AOT_{input}(\lambda)$* is provided by AERONET or CAMS-NRT, and *$AOT_{AM1}(\lambda)$* and *$AOT_{AM2}(\lambda)$* are computed from the two OPAC aerosol models. $\lambda_1$ and $\lambda_2$ are 440 and 870 nm, resp.. The weights $w_{AM1}$ and $w_{AM2}$ are obtained from ***Eq. (12a) and (12b).***, and used at other wavelengths to compute the aerosol transmittance, according to ***Eq. 8.*** For the computation of the diffuse radiation components by SMART-G, the weights $w_{AM1}$ and $w_{AM2}$ are applied to the other aerosol optical properties (phase function, single scattering albedo).

For the sensitivity study of ***Sect. 6***, the AERONET inverted aerosol model provides the aerosol phase function and single scattering albedo at four wavelengths. In this case, *AOT* and the aerosol single scattering albedo are inter and extrapolated at other wavelengths, while the phase function at the closest wavelength is used. The vertical profile of *AOT* varies as an exponential law with a vertical height of 2 km.

## 4. Application of cloud-screening methods based on measured irradiances

The validation is performed in clear-sky conditions, when aerosols affect the surface solar irradiance but not the clouds. This section describes two cloud-screening methods, selected based on the work of Gueymard *et al.* [2019] who compare the outputs of several cloud-screening algorithms, based on time series of irradiance measurements, to cloud cover evaluations from ground-based sky imagers, for several locations in the United States of America. The two methods are expected to show contrasted results in terms of comparison scores, as detailed in ***Sect. 5***.

### 4.1. Choice of the cloud-screening procedure

Since the output of cloud-screening methods is binary, e.g. the sky is either cloudy or clear, Gueymard *et al.* [2019] evaluate the performances of the cloud-screening methods with a confusion matrix. As the aim of our study is to validate SolaRes simulations in clear-sky conditions, we need to select a cloud-screening method that maximizes the number of correctly identified clear-sky cases, or the True Positive score (TPS). It is also important to keep the False Positive score (FPS) as





low as possible to avoid cases of incorrect identification and to minimise cloud contamination. The precision score PS may represent the performance of the screening method in identifying clear-sky moments:


$$PS = \frac{TPS}{TPS + FPS} \tag{13}$$

The cloud-screening algorithm of Garcia *et al.* [2014] (thereafter named Garcia) is retained as it shows the highest PS of 24.0%, and a relatively low FPS of 8.4% [Gueymard *et al.*, 2019]. In
addition, the algorithm of Long and Ackerman [2000] (thereafter named L&A) is retained as it shows the lowest FPS of 7.2 %, with PS of 20.8% [Gueymard *et al.*, 2019], as an alternative with fewer misidentified clear-sky moments.

### 4.2. Description of the chosen cloud-screening procedure

Both Garcia and L&A cloud-screening methods rely on the same series of four tests based on $GHI_{obs}$ and $DifHI_{obs}$ measurements. It's worth mentioning that the Garcia method relies on collocated $AOT$ information, which enables it to better detect the presence of clouds, particularly for higher aerosol loads. The various tests of the Garcia algorithm are adjusted and relaxed to allow the detection of clear-sky moments characterized by higher aerosol loads.

The first two tests remove obvious cloudy moments characterized by extreme values of the normalized global irradiance $GHI_N$ (test 1) and $DifHI_{obs}$ (test 2) through the definition of threshold values. The third and fourth tests can detect more subtle cloud covers by analysing the temporal variability of $GHI_{obs}$ (test 3) and of the normalised diffuse irradiance ratio $D_{R,N}$ defined as the normalised value of $D_{R,obs}$, defined as $DifHI_{obs}$ divided by $GHI_{obs}$ (test 4). Note that the goal of the
normalization step in the first and fourth tests is to lessen the dependency of $GHI_{obs}$ and $DifHI_{obs}$ with respect to $SZA$. The use of such normalized quantities tends to eliminate early morning and late evening events indiscriminately of the cloud cover [Long and Ackerman, 2000]. This behaviour has limited impact in this study as the data set is selected with $SZA$ smaller than 80°.

The four tests are applied in an iterative process to provide each time a new collection of clear-sky
moments on which to fit at a diurnal scale, and a set of daily coefficients $a_{GHI/DR,day}$ and $b_{GHI/DR,day}$:

$$GHI_{obs} = a_{GHI,day}\,\mu_0^{b_{GHI,day}} \tag{14a}$$

$$D_{R,obs} = a_{D_R,day}\,\mu_0^{b_{D_R,day}} \tag{14b}$$

where the two coefficients $a_{GHI,day}$ and $a_{DR,day}$ represent the associated clear-sky global irradiance and diffuse ratio for SZA=0° and the two coefficients $b_{GHI}$ and $b_{DR,day}$ the variations of $GHI$ and $D_R$ with $\mu_0$ for each day, assuming constant $AOT$ during the day. The daily values of each coefficient are then averaged over the available collection of clear-sky days to determine the new annual coefficients $a_{GHI/DR}$ and $b_{GHI/DR}$ over the database, which are then used for the normalization of the measurements in the first and fourth tests. A new set of $a_{GHI/DR}$ and $b_{GHI/DR}$ parameters is determined for each iteration, until convergence is reached within 5%. This method is thus quite versatile and can be applied to any site equipped with measurements of both global and diffuse irradiances.

*Table 1* compares the initial values of the coefficients from Long and Ackerman [2000] and Garcia *et al.* [2014] with the ones found for our study conducted in Lille and Palaiseau over the period





2010-2020. The parameters $GHI_{N,min}$ and $GHI_{N,max}$ correspond to the normalized global irradiance thresholds used in the first test to constrain $GHI_N$. These thresholds are computed as $GHI_{N,\,{max \atop min}} = a_{GHI} \pm 100\,\mathrm{W.m^{-2}}$. The application of the initial L&A method in Lille and Palaiseau produces equivalent scalable parameters $GHI_{N,min}$, $GHI_{N,max}$, $b_{GHI}$ and $b_{DR}$ for both sites.

Garcia *et al.* [2014] modify the L&A method to make it applicable to the particular conditions of the Izana Observatory in the Canary Islands, a high-elevation arid site. They show that the daily mean coefficients $a_{GHI,day}$ and $b_{GHI,day}$ found for that site were somewhat correlated to the variations of $AOT$ measured coincidentally at 500 nm. The variation of $a_{GHI,day}$ with respect to $AOT$ in Lille and Palaiseau was found to be similar to the one used in Garcia *et al.* [2014]. However, the correlation coefficient is only 0.20, which is lower than the value reported by Garcia *et al.* [2014]. Additionally, the correlation coefficient for $b_{GHI}$ is only 0.30, which is significantly smaller than the value of Garcia *et al.* [2014].

In the present study, the variability of the coefficient $b_{DR}$ relatively to $AOT$ is also investigated using various parameterisations. The highest correlation coefficient of 0.31 is found when using a power law of $AOT$. Since this correlation coefficient is close to the one found for $b_{GHI}$, we slightly modify the Garcia method by including the change of $b_{DR}$ with respect to $AOT$ (***Table 1***).

Table 1. Main parameters used by the cloud-screening methods of Long and Ackerman [2000] (L&A) and Garcia *et al.* [2014] (Garcia). It includes the values initially reported in the literature as well as those found specifically for Lille and Palaiseau for the period 2010-2020. *AOT* is the aerosol optical thickness measured at 500 nm.

| Test number | Parameter | Cloud-screening method and source | | | | |
|---|---|---|---|---|---|---|
| | | L&A | | | Garcia | |
| | | Literature | Lille | Palaiseau | Literature | Lille and Palaiseau |
| 1st test | $a_{GHI}$ (W/m²) | / | 1153 | 1140 | $1054 \cdot AOT^{-0.03}$ | |
| | $GHI_{N.min}$ (W/m²) | 1000 | 1053 | 1040 | $1054 \cdot AOT^{-0.03} - 100$ | |
| | $GHI_{N.max}$ (W/m²) | 1250 | 1253 | 1240 | $1054 \cdot AOT^{-0.03} + 100$ | |
| | $b_{GHI}$ | 1.20 | 1.23 | 1.21 | $0.41 \cdot AOT + 1.09$ | $0.17 \cdot AOT + 1.21$ |
| 4th test | $b_{DR}$ | -0.80 | -0.67 | | -0.62 | $-0.54 \cdot AOT^{-0.09}$ |

### 4.3. Impact of the cloud-screening procedures

***Table 2*** shows averaged values of the observed solar resource parameters, under both all-sky and clear-sky conditions, meanwhile ***Table 3*** and ***Fig. 1*** show averaged values of the atmospheric properties observed by AERONET, that are most relevant for radiative transfer simulations of the solar resource components under clear-sky conditions. For the whole paper, *SZA* is constrained below 80°. Winter is composed by December-February, spring by March-May, summer by June-August and autumn by September-November.

A proportion of 14 to 16% of the moments can be declared clear-sky by Garcia in 2018-2019 at Lille and Palaiseau, and only 8 to 10% by the stricter L&A cloud-screening (***Table 2***). The proportion of clear-sky moments in summer is more than twice larger than in winter according to Garcia, and larger by ~35% compared to spring and autumn. L&A also identifies less clear-sky moments in winter but unexpectedly does not show more clear-sky moments in summer than in





spring and autumn. As written hereafter, the results show that L&A also has a tendency to screen-out moments characterised by large *AOT* values which occur more frequently in spring and summer (***Table 3***). Clear-sky (Garcia) contributes by 21.2% to the total accumulated *GHI* at Lille, and by
23.7% at Palaiseau.

The mean solar resource components are quite similar at Lille and Palaiseau, with almost equal *DifHI$_{obs}$* values in both all-sky and clear-sky conditions (***Table 2***), indicating comparable average cloud cover. *DNI$_{obs}$* is larger in Palaiseau than in Lille, with a difference of about 30 W/m² in all-sky conditions, and approximately 20 W/m² in clear-sky conditions. Part of these differences could be
attributed to the smaller mean *SZA* in Palaiseau which is located at a lower latitude than Lille. As a consequence, both all-sky and clear-sky *GHI$_{obs}$* values are around 25 W/m² larger in Palaiseau than in Lille.

Table 2. Averaged solar resource components (*GHI$_{obs}$*, *DNI$_{obs}$*, *DifHI$_{obs}$*) observed in Lille and Palaiseau in 2018-2019, in all-sky and in clear-sky conditions, at 1-minute time resolution (SZA < 80°). The all-sky data set is made by all data, while the clear-sky data set is composed by the minutes identified as cloud-free by either the algorithm of Long and Ackerman [2000] (L&A) or the method of Garcia *et al.* [2014] (Garcia). The second part of the Table gives the number of all-sky minutes, and the proportion (%) of clear-sky minutes, in 2018-2019 and also in function of the season.

|  |  | Lille |  |  | Palaiseau |  |  |
|---|---|---|---|---|---|---|---|
|  | **Time cover** | **All sky** | **Clear sky (L&A)** | **Clear sky (Garcia)** | **All sky** | **Clear sky (L&A)** | **Clear sky (Garcia)** |
| **SZA (°)** | **2018–2019 mean ± standard deviation** | 59 ± 15 | 60 ± 14 | 58 ± 15 | 58 ± 15 | 58 ± 14 | 57 ± 15 |
| **GHI$_{obs}$ (W/m²)** |  | 330 ± 252 | 474 ± 218 | 493 ± 229 | 352 ± 264 | 500 ± 222 | 516 ± 227 |
| **DNI$_{obs}$ (W/m²)** |  | 303 ± 341 | 765 ± 132 | 739 ± 144 | 333 ± 350 | 784 ± 124 | 758 ± 139 |
| **DifHI$_{obs}$ (W/m²)** |  | 162 ± 108 | 79 ± 22 | 92 ± 35 | 160 ± 107 | 79 ± 23 | 93 ± 33 |
| **Number of all-sky minutes, and proportion of clear-sky minutes (%)** | **2018-2019** | 379 717 | 7.8% | 14.2% | 427 480 | 9.8% | 16.2% |
|  | **Winter** | 50 446 | 6.9% | 8.3% | 67 769 | 7.4% | 8.9% |
|  | **Spring** | 112 195 | 7.8% | 13.0% | 125 242 | 7.9% | 13.9% |
|  | **Summer** | 133 665 | 7.8% | 17.9% | 142 373 | 10.5% | 20.5% |
|  | **Autumn** | 83 411 | 8.7% | 13.3% | 92 096 | 12.9% | 17.9% |

The cloud-screening methods agree to show a strong impact in *GHI$_{obs}$*, *DNI$_{obs}$* and *DifHI$_{obs}$*, compared to all-sky conditions. The influence of the chosen cloud-screening method is more important in *DNI$_{obs}$* and *DifHI$_{obs}$* than in *GHI$_{obs}$*. Indeed, in clear-sky conditions, *DifHI$_{obs}$* is divided by a factor of 1.7-2.0 at Lille, *DNI$_{obs}$* is multiplied by a factor of 2.3-2.5, and *GHI$_{obs}$* is multiplied by a factor of ~1.45. Both cloud-screening methods have a comparable impact in *DNI$_{obs}$*, which
increases by 420-450 W/m² at both locations. *DifHI$_{obs}$* in clear-sky conditions at Lille decreases by





83 W/m² with L&A, compared to all-sky, and by 70 W/m² with Garcia. The Garcia cloud-screening then keeps more scattering than L&A, either caused by aerosols either by unfiltered clouds. It is interesting to note that the standard deviation in $DifHI_{obs}$ also strongly decreases from 67% in all-sky conditions at Lille (compared to the average) to 38% with Garcia clear-sky, and 28% with L&A

clear-sky, and in $DNI_{obs}$ from 113% in all-sky to 19% in Garcia clear-sky and to 17% in L&A clear-sky. L&A cloud-screening increases $GHI_{obs}$ by ~145 W/m² while Garcia cloud-screening increases $GHI_{obs}$ by ~160 W/m² at both Lille and Palaiseau.

*Table 3* presents mean *AOT*, Ångström exponent and water vapour content (WVC) measured by AERONET in Lille and Palaiseau in 2018-2019, according to the two cloud-screening methods, and

*Fig. 1* shows the seasonal dependence of *AOT* and *WVC*. The clear-sun data set is composed by the AERONET Level 2.0 data set, which screens out measurements with clouds detected in the only solar direction. The other two data sets are made by combining the Level 2.0 AERONET data cloud-screening and one of the two irradiance cloud-screening methods. Hence in the latter case, only cloud-free irradiance data points coincident with Level 2.0 AERONET measurements are

considered.

Table 3. Average and standard deviation of instantaneous atmospheric properties measured at Lille and Palaiseau by AERONET in 2018-2019: AOT at 550 nm, the Ångström exponent, and the water vapour column content (WVC). In clear sun conditions, the number of observations represents the

total number of Level 2.0 AERONET measurements while in clear-sky it corresponds to the number of minutes identified as cloud-free by either the algorithm of Long and Ackerman [2000] (L&A) or the method of Garcia *et al.* [2014] (Garcia), coincident to the Level 2.0 AERONET data.

| | Lille | | | Palaiseau | | |
|---|---|---|---|---|---|---|
| | **Clear sun (Level 2.0)** | **Clear sun & sky (Level 2.0 + L&A)** | **Clear-sun & sky (Level 2.0 + Garcia)** | **Clear sun (Level 2.0)** | **Clear-sun & sky (Level 2.0 + L&A)** | **Clear-sun & sky (Level 2.0 + Garcia)** |
| **Number of obs.** | 25 739 | 7 501 | 13 189 | 26 294 | 9 757 | 16 156 |
| **AOT at 550 nm** | 0.14 ± 0.10 | 0.10 ± 0.05 | 0.13 ± 0.08 | 0.13 ± 0.08 | 0.08 ± 0.04 | 0.11 ± 0.07 |
| **Ångström Exponent** | 1.29 ± 0.40 | 1.34 ± 0.32 | 1.34 ± 0.36 | 1.30 ± 0.38 | 1.30 ± 0.32 | 1.31 ± 0.35 |
| **WVC (cm)** | 1.5 ± 0.7 | 1.4 ± 0.5 | 1.6 ± 0.6 | 1.6 ± 0.7 | 1.4 ± 0.5 | 1.6 ± 0.6 |

The clear-sun data set shows that the aerosol properties and WVC are highly variable in Lille and

Palaiseau. The standard deviation is 71% in *AOT* at 550 nm at Lille, 31% in the Ångström exponent, and 47% in the *WVC* (*Table 3*). Significant part of this variability is explained by seasonal influence, as mean *AOT* increases by a factor of 1.8 from winter to spring, and *WVC* increases by a factor of 3 from winter to summer. Variability can also occur within the season as between two consecutive days. Indeed the standard deviation in *AOT* in spring remains close to the

standard deviation over a year. The 90th percentile of the AOT distribution at Lille is 0.32, and *AOT* could even be larger than 0.80 as on 2018/06/06 and 2019/03/31. The intra-seasonal variability is less important in *WVC* as the standard deviation in summer falls down to 24%.





The differences between Lille and Palaiseau are small, in terms of mean values and variability of the atmospheric properties that are most relevant for clear-sky radiative transfer simulations, consistently with Ningombam *et al.* [2019], for the time period 1995-2018.

The clear-sky conditions using the Garcia cloud-screening method are more representative of the *AOT* variability than those detected with the L&A method:

- The number of clear-sky minutes is larger in the Garcia than in the L&A data set (**Table 3**).

- The annual means and standard deviations of *AOT* by the Garcia cloud-screening method are closer to the clear-sun values than those obtained by the L&A method, and especially in spring-summer when L&A significantly under estimates the clear-sun means (**Fig. 1**).

- The relative increase of mean *AOT* from winter to spring by Garcia was equal to the increase during clear-sun conditions, while the increase was less intense under L&A conditions (**Fig. 1**).

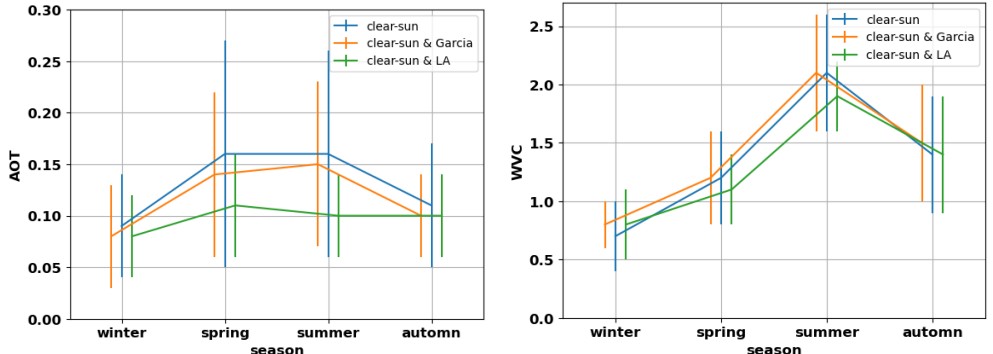

Figure 1. Seasonal dependence of AOT and WVC (cm) at Lille in 2018-2019, according to Level 2.0 AERONET, and for two cloud-screening methods. Error bars show the standard deviation for each season.

## 5. Validation with AERONET as input data

SolaRes computations of solar resource standard components of *GHI*, *DNI*, and *DifHI* are compared to ground-based measurements made at Lille and Palaiseau in 2018-2019, at the 1-minute time resolution. Furthermore, SolaRes computations are also compared to ground-based measurements of GTI at Lille in 2019. AERONET provides the input spectral *AOT*, which is averaged at the 15-minute time resolution. The continental clean and desert dust OPAC models are mixed to reproduce AERONET spectral *AOT* (**Sect. 3.3**). AERONET also provides observed *WVC*, and AERONET V3 provides the ozone column content. Daily averages of surface albedo delivered by the CAMS-radiation service are used.

Comparison scores are showed and commented in this section, which are the relative mean bias difference (MBD) and the relative root mean square difference (RMSD), which are usual indicators of dispersion, as commented by Gueymard [2014], and used by many authors [e.g. Ruiz-Arias *et al.*, 2013; Sun *et al.* 2019]:



$$MBD = \frac{100}{obs_{mean}} \frac{\sum_{i=1}^{N}(comp_i - obs_i)}{nb} \quad,$$
(15a)

$$RMSD = \frac{100}{obs_{mean}} \left[\frac{\sum_{i=1}^{N}(comp_i - obs_i)^2}{nb}\right]^{1/2} \quad,$$
(15b)

where obs stands for the observed quantity, and comp for the computation by SolaRes, which can be *GHI, DNI, DifHI, DifTI*. The sum is made over the pair number nb. $obs_{mean}$ stands for the averaged observed quantity, and the factor 100 provides *MBD* and *RMSD* in %. Best agreement is reached if the values of the comparison scores are zero.

At Lille in 2018-2019, 8500 radiative transfer computations of *DifHI* are performed at the 15-minute time resolution. SolaRes then provides solar resource components for 183 000 1-minute time steps in clear-sun conditions. Only data within a temporal window of ±10 minutes around the AERONET record time is kept, and the SolaRes data set then reduces to 125 000 time steps. A further screening is applied on *SZA*, keeping values smaller than 80°, as done by e.g. Ruiz-Arias *et al.* [2013]. Comparison data pairs are generated by associating coincident simulation and observation at 1-minute time resolution. Eventually, the cloud-screening procedures on solar irradiance measurements (S*ect. 4*) are applied to keep clear-sky conditions. At Lille in 2018-2019, 50 000 comparison data pairs are constituted with the Garcia cloud-screening procedure (which represents 13.2% of all-sky data, only 1% less than the cloud-screened data set by the only irradiance measurements, see *Table 2*) and 26 000 comparison data pairs with the L&A cloud-screening procedure (*Table 4*). Slightly more AERONET data are available for radiative transfer computations at Palaiseau over the same years, and more comparison pairs are eventually kept, as ~65 000 with the Garcia cloud-screening and 37 000 with the L&A cloud-screening.

First, comparisons scores in *GHI* are presented in *Sect. 5.1*, then comparison scores in both *DNI* and *DifHI*, without (*Sect. 5.2*) and with the circumsolar contribution (*Sect. 5.3*).

### 5.1. GHI at Lille and Palaiseau

As described in *Sect. 2.1*, $GHI_{obs}$ is measured by four Kipp&Zonen instruments at both Lille and Palaiseau. $GHI_{obs}$ is obtained by summing $DirHI_{obs}$ and $DifHI_{obs}$ (*Eq. (1)*), measured by a CHP1 pyrheliometer and a shaded CMP22 pyranometer, respectively, and also measured at Lille by a CMP11 pyranometer during a time period extending over part of spring and summer 2018.

*Table 4* and *Figure 2* present the comparison scores in *GHI*. The correlation coefficient between $GHI_{obs}$ and $GHI_{RT}$ was 0.999 for the two sites (not shown in *Table 4*). With the Garcia cloud-screening, $GHI_{obs}$ is slightly underestimated, by 0.4% (Palaiseau) to 0.8% (Lille). The absolute under-estimation is -3.8±8.1 W/m$^2$ at Lille, with 55% of 1-minute values included between -5 and 5 W/m$^2$, which is of the order of the 5 W/m$^2$ uncertainty requirement for the measurements by BSRN [Ohmura *et al.*, 1998]. The *RMSD* in *GHI* is around 1.6% at both Lille and Palaiseau, with the Garcia cloud-screening.





Table 4. Comparison scores in *GHI*, at both Lille and Palaiseau, for two cloud-screening procedures (Garcia and L&A as described in **Sect. 4**), over different time periods: whole year in 2018-2019, on different seasons, and spring and summer 2018 by the CMP11. The number of comparison pairs (1-minute resolution), the averaged $GHI_{obs}$, as well as *MBD* and *RMSD* (**Eq. (15)**) are given.

| Location | Instruments | Time period | cloud-screening | Number of comparison pairs | Mean $GHI_{obs}$ (W/m²) | Comparison scores | |
|---|---|---|---|---|---|---|---|
| | | | | | | MBD (%) | RMSD (%) |
| Lille | CH1+CMP22 | All seasons | Garcia | 50 000 | 500±228 | -0.8 | 1.7 |
| | | All seasons | L&A | 26 000 | 482±218 | -0.5 | 1.2 |
| | | Winter/spring/summer/autumn | Garcia | 3 900 / 13 500 / 22 800 / 9 800 | 324 / 531 / 552 / 409 | -0.7/-1.3 / -0.8 / -0.1 | 1.5 / 1.9 / 1.6/1.6 |
| | CMP11 | Part of spring+summer 2018 | Garcia | 7450 | 538±234 | -0.0 | 2.2 |
| Palaiseau | CMP22 | All seasons | Garcia | 65 400 | 517±227 | -0.4 | 1.5 |
| | | All seasons | L&A | 37 500 | 503±219 | -0.1 | 1.0 |

The comparison with the CMP11 shows a better score in *MBD* and a worst score in *RMSD*. The worst score in *RMSD* is explained by the seasonal influence, studied with the CHP1+CMP22 comparison scores. Worst agreement is observed in spring, with a *MBD* of -1.3% and a *RMSD* of 1.9%, which is close to the *RMSD* of 2.2% with the CMP11 in spring-summer. These values of *RMSD* are similar to the *RMSD* of 1.9% between the observations themselves (**Sect. 2.1**). The better
score in *MBD* with CMP11 than with CHP1+CMP22 may be explained by the difference between the observations themselves. Indeed according to the computations, the influence of the CMP11 spectral bandwidth in $GHI_{RT}$ is 4.5±2.5 W/m², or 0.8±0.3%, which is significantly smaller than the observed difference of 1.6% between CMP11 and CHP1+CMP22 $GHI_{obs}$ (**Sect. 2.1**). The cosine error of the unshaded CMP11 pyranometer may be responsible for this discrepancy. Consequently,
the agreement between SolaRes and observations is improved with the CMP11 data set, in terms of *MBD*.

The cloud-screening method has a significant impact on the comparison scores. For example on 20 April 2018 at Lille, largest disagreement reaching 60 W/m² is observed during the afternoon between the Garcia data set and the simulation (**Fig. 3**). This is certainly caused by clouds in the sky
vault but undetected by the Garcia cloud-screening as 1) the L&A screening procedure gets rid of these points, consistently with its lower FPS by Gueymard *et al.* [2019], and 2) AERONET Level 2.0 provides values of *AOT* and *WVC* all day, meaning that no clouds are seen in the solar direction. Such a cloud cover has less impact after 16:00 when agreement improves.



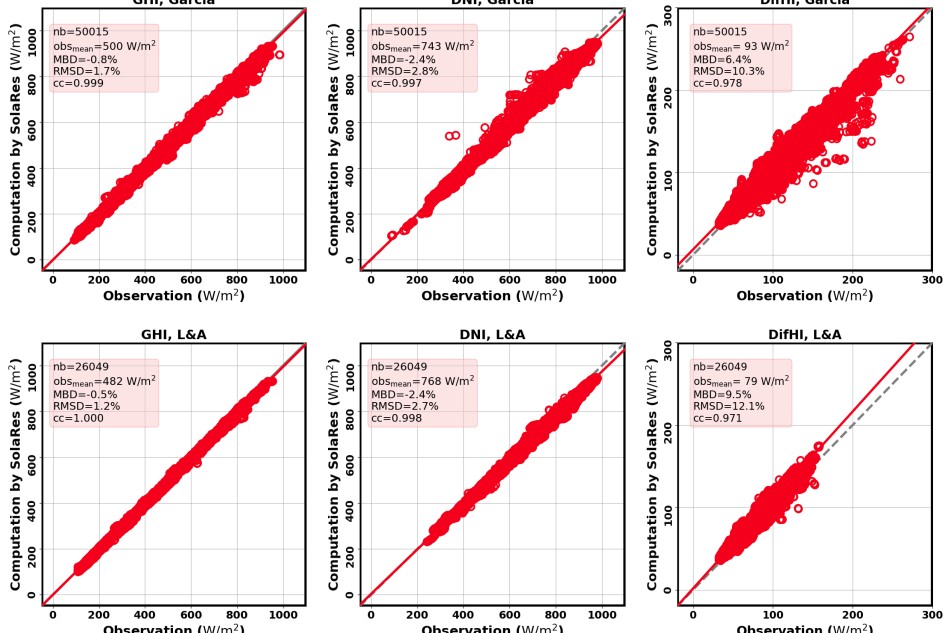

Figure 2. Comparison between 1-minute computations and observations at Lille in 2018-2019 (by CHP1+CMP22) in clear-sky conditions, for *GHI* (left), *DNI* (centre), and *DifHI* (right). Clear-sky was defined by the Garcia cloud-screening method (top) and the L&A method (bottom), with *SZA* < 80°, and within 10 minutes of AERONET record time of *AOT*. MBD and RMSD are given according to ***Eq. 15***, *nb* is the pair number, $obs_{mean}$ is the mean value of the observed parameter, and cc is the correlation coefficient of the linear interpolation (red line). The dashed grey line is the 'x=y' line.

Such a behaviour has as consequences that the mean comparison scores over the full time period are improved with the L&A cloud-screening procedure (***Table 4 and Fig. 2***). The L&A cloud-screening procedure decreases *MBD* in *GHI* by ~0.3%, and *RMSD* by ~0.5%. *MBD* could be as low as -0.1% at Palaiseau with L&A, 64% of the *MBD* values lying within ±5 W/m² of GHIobs. *RMSD* could be as low as 1.0%, confirming the success of the radiative closure study involving pyranometers, AERONET *AOT* and SolaRes, equally to results showed by Ruiz-Arias *et al.* [2013] but with AERONET-inverted products.



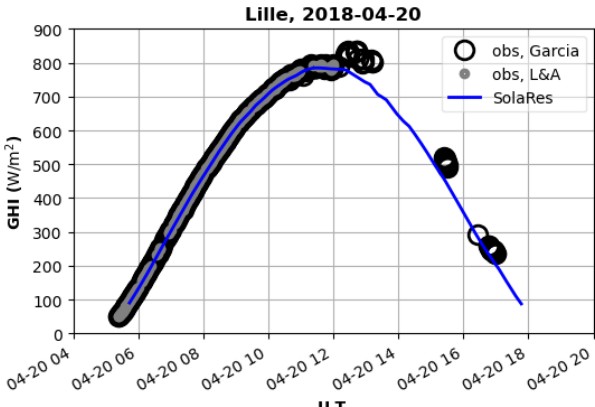

Figure 3. Global horizontal irradiance (GHI) observed at Lille on 2018/04/20 in clear-sky conditions, and simulated by SolaRes. $GHI_{obs}$ is cloud-screened by both Garcia and L&A methods.


### 5.2. DNI and DifHI without the circumsolar contribution

Both $DNI_{obs}$ and $DifHI_{obs}$ are separately measured at Lille and Palaiseau by the CHP1 pyrheliometer and the shaded CMP22 pyranometer, resp.. **Tables 5 and 6** present the comparison scores for *DNI* and *DifHI*, resp., as well as **Fig. 2**. In this section, the circumsolar contribution is not computed, $DNI_{strict}$ is compared to $DNI_{obs}$, and $DifHI_{strict}$ to $DifHI_{obs}$.


$DNI_{strict}$ is under estimated by -1.6% at Palaiseau and -2.4% at Lille (**Table 5 and Fig. 2**) with the Garcia cloud-screening method, and *RMSD* is 2.2% at Palaiseau and 2.8% at Lille. These results are highly satisfactory given the 5% uncertainty claimed by Gueymard and Ruiz-Arias [2015] for uncertainty of 0.02 in *AOT*, as that of AERONET measurements.

As expected, the dependence on the cloud-screening procedure is little, as the criteria on direct solar irradiance are similar between the two cloud-screening procedures. Similar values in *MBD* and RMSD (in %) show that the performance is stable whatever the *AOT* range, the L&A cloud-screening generating a smaller data set missing *AOT* variability, compared to Garcia. We can confidently guess negligible residual cloud influence as AERONET screens out clouds in the solar direction in the Level 2.0 quality, and it is associated with the solar irradiance cloud-screening methods.


While $DNI_{strict}$ is under estimated, $DifHI_{strict}$ is over estimated, by 5-6% at Lille and Palaiseau with the Garcia cloud-screening (**Table 6 and Fig. 2**). According to **Eq. 9** and **10**, both $DNI_{obs}$ under-estimation and $DifHI_{obs}$ over-estimation are expected, as the circumsolar contribution is not considered here.


*RMSD* in *DifHI* is ~10%, which is significantly larger than *RMSD* in both *GHI* and *DNI*. It is expected as 1) *DifHI* depends on the distinction between scattering and absorption, while $DNI_{strict}$ depends only on extinction; 2) moreover *DifHI* depends on surface reflection while $DNI_{strict}$ depends only on atmospheric extinction. The better agreement in *GHI* (**Sect. 5.1**) than in both *DNI* and *DifHI* shows that *MBD* in both *DNI* and *DifHI* mostly compensates.




Table 5. As **Table 4**, but for *DNI_obs* measured by the CHP1 pyrheliometer.

| Location | Time period | cloud-screening | Circumsolar contribution simulated | Comparison pair numbers | Mean $DNI_{obs}$ (W/m²) | Comparison scores MBD (%) | RMSD (%) |
|---|---|---|---|---|---|---|---|
| Lille | Whole year | Garcia | no | 50 000 | 743±141 | -2.4 | 2.8 |
| | Whole year | L&A | no | 26 000 | 768±120 | -2.4 | 2.7 |
| | Whole year | Garcia | yes | 50 000 | 743±141 | -1.2 | 2.2 |
| | Winter/spring/summer/autumn | Garcia | no | 3 900 / 13 500 / 22 800 / 9 800 | 742 / 757 / 737 / 737 | -2.0 / -2.5 / -2.5 / -2.4 | 2.6 / 2.8 / 2.8 / 2.9 |
| Palaiseau | Whole year | Garcia | no | 65 400 | 758±139 | -1.6 | 2.2 |
| | Whole year | L&A | no | 37 500 | 785±123 | -1.6 | 1.8 |
| | Whole year | Garcia | yes | 65 400 | 758±139 | -0.5 | 1.8 |


It may be surprising that *MBD* in *DifHI* increases with the L&A cloud-screening procedure. This is partly caused by the significant decrease in mean *DifHI*, as L&A screens out atmospheric conditions with largest diffuse irradiance cases. Similarly, *MBD* is significantly smaller in spring-summer than in autumn-winter, partly because mean *DifHI* is larger.

Both mean $GHI_{obs}$ and mean $DirHI_{obs}$ are much larger at Palaiseau according to Gschwind *et al.* [2019] than with our cloud-screening procedures: $GHI_{obs}$ averaged over 2005-2007 is 600 W/m², and mean $DirHI_{obs}$ is 492 W/m² with a strict cloud-screening procedure keeping only ~10 000 data 1-minute data per year. Consequently, $DifHI_{obs}$ is 108 W/m² for Gschwind *et al.* [2019], also larger than with our cloud-screening procedures. According to **Table 2**, $DirHI_{obs}$ is ~420 W/m² at Palaiseau, subtracting $DifHI_{obs}$ to $GHI_{obs}$. It must be noted that mean solar resource parameters remain unchanged at Palaiseau (**Table 2**) when adding the AERONET cloud-screening (**Table 4**).

Table 6. As **Table 4**, but for *DifHI_obs*, measured by the CMP22 pyranometer in 2018-2019.

| Location | Time period | cloud-screening | Circumsolar contribution simulated | Comparison pair number | Mean $DifHI_{obs}$ (W/m²) | Comparison scores MBD (%) | RMSD (%) |
|---|---|---|---|---|---|---|---|
| Lille | Whole year | Garcia | no | 50 000 | 93±35 | 6.4 | 10.3 |
| | Whole year | L&A | no | 26 000 | 79±22 | 9.5 | 12.1 |
| | Whole year | Garcia | yes | 50 000 | 93±35 | 2.4 | 9.4 |
| | Winter/spring/summer/autumn | Garcia | no | 3 900 / 13 500 / 22 800 / 9 800 | 62 / 99 / 102 / 77 | 7.0 / 5.6 / 6.4 / 7.5 | 9.4 / 9.8 / 10.2 / 11.1 |
| Palaiseau | Whole year | Garcia | no | 65 400 | 92±33 | 5.1 | 10.0 |
| | Whole year | L&A | no | 37 500 | 80±23 | 7.5 | 10.0 |
| | Whole year | Garcia | yes | 65 400 | 92±33 | 1.3 | 9.3 |


As showed in **Sect. 4**, when the cloud-screening is stricter, atmospheric scattering is reduced, and $DifHI_{obs}$ may decrease and $DNI_{obs}$ on contrary may increase. As the Gschwind *et al.* [2019] cloud-screening increases both $DifHI_{obs}$ and $DirHI_{obs}$, atmospheric scattering is not in play. The other





important factor is *SZA*. We could then make the hypothesis that the Gschwind *et al.* [2019] cloud-screening procedure rejects large values of *SZA*, and mean *SZA* would be smaller than in our data set (**Table 2**), explaining the increase in both *DirHI$_{obs}$* and *DifHI$_{obs}$* and consequently in *GHI$_{obs}$*.

According to **Table 4**, the latitude influence is ~15 W/m$^2$ in *GHI$_{obs}$* between Lille and Palaiseau, and the cloud-screening influence is also ~15 W/m$^2$.

### 5.3. DNI and DifHI with the circumsolar contribution

In this Section, we consider *DNI$_{pyr}$* and *DifHI$_{pyr}$*, which are corrected by the circumsolar contribution to better represent the measurements, according to **Eq. 9** and **10**. The circumsolar contribution to the direct normal radiation, *ΔDifNI$_{circ}$*, is 8±6 W/m$^2$ on average (similar on both sites), with a median and a 90$^{th}$ percentile of 6 and 15 W/m$^2$, resp. *ΔDifNI$_{circ}$* then represents 1.2±1.3% of *DNI$_{strict}$*, with a 675 median of 0.7%, and a 90$^{th}$ percentile of 2.4%. **Figure 4** shows *ΔDifNI$_{circ}$* in function of both the Ångström exponent α and the slant aerosol optical thickness at 550 nm (SOT) which is defined as *AOT* divided by μ$_0$ [Blanc *et al.*, 2014]. Most values of *ΔDifNI$_{circ}$* are smaller than 20 W/m$^2$, consistently with simulations by Blanc *et al.* [2014]. Values larger than 20 W/m$^2$ mostly occurs for small α and/or large *SOT*.


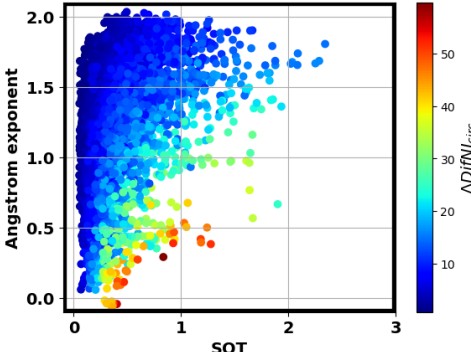

Fig. 4. *ΔDifNI$_{circ}$* in function of both the Ångström exponent and the slant path optical thickness at 550 nm (SOT).

Adding *ΔDifNI$_{circ}$* to *DNI$_{strict}$* improves the comparison scores: *MBD* in *DNI$_{pyr}$* decreases by more than 1%, and *RMSD* by ~0.5%. Under estimation should be expected when circumsolar contribution is not considered, meaning that the excellent results by Ruiz-Arias *et al.* [2013] with *DNI$_{strict}$* could indicate on contrary over estimation of *DNI$_{obs}$* by *DNI$_{pyr}$*.

The mean circumsolar contribution to diffuse horizontal, ΔDifHI$_{circ}$, is 4±2 W/m$^2$, and the 690 comparison scores with DifHI$_{pyr}$ also significantly improves, with MBD decreasing by more than 4% and RMSD slightly decreasing by less than 1%.




### 5.4. Diffuse irradiance in a vertical plane

#### 5.4.1. Two regimes

$GTI_{obs}$ is measured by the CMP11 pyranometer at Lille from 2019/01/18 to 2019/12/31, the instrument being tilted vertically at 90°, and oriented at an azimuth angle of 180°, i.e. facing the South direction. Signal in summer shows two distinct regimes, as for example on 27 June 2019 (**Fig. 5.4**):

1. Most of the day around noon, the Sun in the southern half-sky faces the instrument, and is then included in the instrument field of view.

2. At both beginning and end of the day, the Sun could be behind the instrument in the northern half-sky, the instrument sensor then being in shadows.

In the second regime, only diffuse radiation is observed while in the first regime, both diffuse and direct radiation contribute to the observed signal. Comparisons are made in both regimes independently.


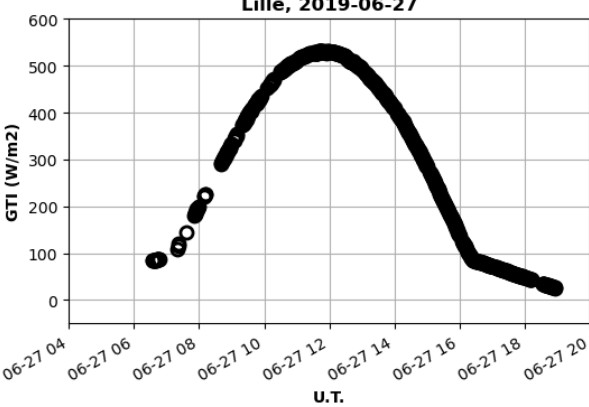

Figure 5. Global tilted irradiance ($GTI_{obs}$) observed by the CMP11 pyranometer in a vertical plane facing South, on 2019/06/27 at Lille. The sun is southwards between 07:14 and 16:27.

#### 5.4.2. Diffuse contribution at both beginning and end of the day in summer

The Sun passing in the northern half-sky, the observed radiation changes of regime. The observed radiation becomes less dependent on $SZA$, generating the flatter wings at the end of the day than around noon in **Fig. 5**.

Comparison in $GTI$ is made selecting SAA larger than 270° (end of the day in summer). Around a
thousand comparison pairs are generated. Observation is over estimated by 6% and the $RMSD$ is 8.5% (1st line in **Table 7**). By selecting $SAA$ smaller than 90° (beginning of the day), the over estimation is 8.7% and the $RMSD$ is 12.1%. These results are similar to the comparison scores in $DifHI$ (**Table 6**).






Table 7. As **Table 4** but for *GTI* in the vertical plane facing South at Lille in 2019, with the Garcia cloud-screening procedure. The time period is defined by season and by the range of *SAA*. Computations are also made for different values of the surface albedo.

| Time period | Surface albedo | Comparison scores | | |
|---|---|---|---|---|
| | | Number of comparison pairs | MBD (%) | RMSD (%) |
| SAA > 270° (only summer) | 0.13 | 1109 | 6.0 | 8.5 |
| 90 < SAA < 270° | 0.13 | 18 655 | -0.6 | 5.0 |
| 90 < SAA < 270°, summer | 0.13 | 9395 | 3.7 | 4.9 |
| 90 < SAA < 270°, winter | 0.13 | 2654 | -6.5 | 6.8 |
| 90 < SAA < 270°, winter | 0.35 | 2654 | -0.2 | 1.4 |

*5.4.3. The influence of changing surface albedo on GTI*

Comparison for the Sun facing the instrument (90° < *SAA* < 270°) showed that $GTI_{obs}$ can be reproduced but with a *RMSD* of 5% (2nd line in **Table 7**). The *RMSD* larger in *GTI* than in *GHI* (**Table 4**) is partly caused by variability in the effective surface albedo.

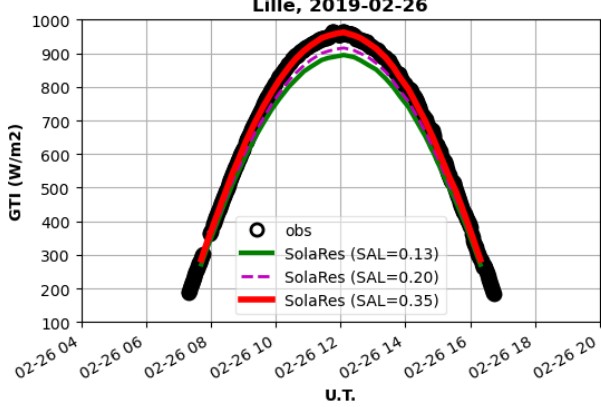


Figure 6. As **Fig. 5** but for 26/02/2019, and with SolaRes estimates for different values of the surface albedo (SAL). According to MODIS, the daily average of the surface albedo is 0.13.

By distinguishing winter and summer seasons, *MBD* changes from +3.7% in summer to -6.5% in
winter (3rd and 4th lines in **Table 7**). While the surface albedo derived from satellite changes little, computations for 26 February 2019 shows that observations can be reproduced with an effective surface albedo of 0.35 (**Fig. 6**), explaining the under estimation of 6.5%. The under estimation then decreases from 6.5% to 0.2%, and RMSD reaches 1.4% (5th line in **Table 7**), which is similar to results in GHI (**Table 4**). Heterogeneities in the albedo of building's walls at local scale, and
subsequent 3D effects, could be responsible of such differences between a satellite surface albedo and an effective surface albedo for a vertical instrument. The differences between winter and summer could be caused by fallen leaves of surrounding trees, in relation with the sun position in the sky.



## 6. Influence of the aerosol parameterisation and the data source

*DNI* is computed by modelling *AOT* at wavelengths describing the solar spectrum, and by the spectral integration of *Eq. 6*. Radiative transfer computations of *DifHI* necessitate not only *AOT* at the same wavelengths, but also the aerosol phase function and the aerosol single scattering albedo. These parameters could hardly be provided by observation, and measurements partially describe such aerosol optical properties. It is therefore necessary to employ various strategies to get the necessary parameters from observation data sets. For example the measured data set can be inverted to provide a fully-described microphysical aerosol model, assuming some hypotheses, which is then usable in radiative transfer computations. AERONET provides such inverted aerosol models, but with a time resolution smaller than the *AOT* time resolution.

Given the high time variability of aerosol properties, the time resolution is an important factor in solar resource estimation [e.g. Sun *et al.*, 2019], and we choose in this paper to rely on Level 2.0 AERONET *AOT* acquired at around 3 minute resolution, when the time resolution of the inverted aerosol model could be ~1 hour. Also, we choose in SolaRes to derive aerosol optical properties by mixing two OPAC aerosol models in such proportions that they reproduce *AOT* measured at two wavelengths (*Sect. 3*). This Section shows the sensitivity of the computed solar resource parameters to the parameterisation of the aerosol properties and also to the aerosol data source.

First, the OPAC aerosol models are modified to show their influence (*Sect. 6.1*). We also show the best results which could be obtained with SolaRes in clear-sky conditions by exploiting inverted aerosol models provided by AERONET (*Sect. 6.2*), instead of the parameterisation reproducing spectral *AOT*. The source of input data is also changed from the AERONET site-defined data set to the CAMS-NRT regular-grid global data set (*Sect. 6.3*) to evaluate the uncertainty in the global mode.

### 6.1. Impact of the aerosol parameterisation: the aerosol model combination

Atmospheric optical properties are necessary input data of a radiative transfer code. In clear-sky conditions, aerosols are the main source of variability of the atmospheric optical properties. Necessary aerosol optical properties are the optical thickness, the phase function and the single scattering albedo at any wavelengths. Measurements are exploited to reproduce the temporal variability in aerosol optical properties. However, measurements can rarely provide all necessary optical properties, as the full phase function and the single scattering albedo. Consequently, we usually need a parameterisation which relates observations to necessary aerosol optical properties. In our case, we use *AOT* at the two wavelengths of 440 and 870 nm to constrain the mean aerosol burden and also as an indicator of the mean aerosol size. Two aerosol OPAC models are mixed in order to reproduce the observed *AOT* (*Eq. 12*). This mixture defines aerosol microphysical properties (size distribution and refractive index) which are processed according to Mie theory to provide the aerosol optical properties as the phase function and the single scattering albedo at any wavelengths.

Validation in *Sect. 5* is performed with a mixture of continental clean and desert dust aerosols. The aerosol OPAC models are changed here to show the sensitivity of the solar resource parameters on the aerosol parameterisation. To best reproduce the observed *AOT* spectral variability, an aerosol model mainly composed by relatively small aerosols is mixed with an aerosol model composed by larger aerosols. The small aerosol models are named by OPAC as continental clean, continental polluted, and urban, and the larger aerosol models are named desert dust, maritime clean, maritime polluted. *Table 8* shows the impact of several aerosol model combinations on the comparison scores





between observation and simulation. The Garcia cloud-screening is used on observation made in 2018 at Lille, and circumsolar contribution is considered.

$DNI_{pyr}$ is the least sensitive parameter, with *MBD* changing between -1.3 to -1.7%, and *RMSD* remaining around 2.5% (**Table 8**). As only the circumsolar contribution in $DNI_{pyr}$ depends on the angular scattering and on the absorption rate of solar radiation, the sensitivity is mainly caused by

the spectral behaviour of *AOT*. $DifHI_{pyr}$ does however depend on both the phase function and the single scattering albedo, and becomes much more dependent on the aerosol models than $DNI_{pyr}$. Absorption coefficient increases from continental clean to continental polluted and to the urban model, consequently $DifHI_{pyr}$ decrease, and also *MBD* from ~+3% to ~-12%, with the model for larger aerosols (**Table 8**) having a secondary influence.


Table 8. Sensitivity of the solar resource components to the OPAC aerosol models, in terms of *MBD* and *RMSD* in *GHI*, $DNI_{pyr}$, and $DifHI_{pyr}$. *cc* stands for continental clean, *cp* for continental polluted and *ur* for urban. *dd* stands for desert dust, *mc* for maritime clean and *mp* for maritime polluted. Computations are made with observation made in 2018 at Lille, using the Garcia cloud-screening.

| Aerosol models | GHI | | $DNI_{pyr}$ | | $DifHI_{pyr}$ | |
|---|---|---|---|---|---|---|
| | MBD (%) | RMSD (%) | MBD (%) | RMSD (%) | MBD (%) | RMSD (%) |
| cc_dd | -0.7 | 1.8 | -1.0 | 2.4 | 2.2 | 10.3 |
| cp_dd | -2.2 | 3.0 | -1.6 | 2.5 | -4.1 | 12.3 |
| ur_dd | -3.7 | 4.7 | -1.7 | 2.5 | -12.3 | 19.9 |
| cc_mc | -0.7 | 1.8 | -1.3 | 2.4 | 3.1 | 10.4 |
| ur_mc | -3.6 | 4.9 | -1.7 | 2.5 | -11.7 | 20.6 |
| cc_mp | -0.6 | 1.7 | -1.3 | 2.4 | 3.3 | 10.4 |
| ur_mp | -3.3 | 4.1 | -1.8 | 2.5 | -8.4 | 16.4 |


The impact on *GHI* is significant, mainly because of the sensitivity of $DifHI_{pyr}$ to the small aerosol model. The efficient compensation between $DNI_{pyr}$ under estimation and $DifHI_{pyr}$ over estimation mostly occurs with the continental clean (cc) model, then providing the best scores in *GHI*, with a *MBD* of -0.7% and a *RMSD* of 1.8% in 2018 at Lille. This is consistent with the large value of

averaged *SSA* at Lille in 2018, as inverted from AERONET measurements (**Sect. 2**). The choice of the larger aerosol model has little influence. No combination could change the sign of *MBD* in *GHI* to positive.

### 6.2. Impact of the aerosol parameterisation: the AERONET-inverted aerosol optical properties as
data source instead of spectral AOT

AERONET provides not only *AOT* measurements at several wavelengths but also the inverted aerosol models [Dubovik *et al.*, 2000; 2002], which can be used as input data by SolaRes. The aerosol phase function and single scattering albedo provided at 4 wavelengths by AERONET at Lille in 2018 are used. As the Level 2.0 data set is too sparse, we choose to use the Level 1.5 data

quality, with possible inconvenience on solar resource precision. The time resolution is around 1 hour, and 420 time steps are available in 2018 at Lille, instead of the ~13 000 Level 2.0 *AOT* time steps. The ±10 minute condition applied on the *AOT* data set is not applied here, in order to get as many 1-minute data pairs as possible.





**Table 9** shows the comparison scores for *GHI*, *DNI$_{pyr}$* and *DifHI$_{pyr}$*. The *RMSD* in *GHI* decreases
from 1.7 to 1.2% with Garcia, and from 1.2 to 0.8% with L&A, while *MBD* reaches 0. Ruiz-Arias
*et al.* [2013] also make comparisons between observation and computations exploiting Level 1.5
AERONET inverted products with a radiative transfer code, but for smaller mean *AOT*. In *GHI*, our
performances are similar with *RMSD* of ~1% and *MBD* of 0%. Such high performance is also
attained with the AERONET *AOT* data set at Palaiseau, and the L&A cloud-screening method. We
demonstrate the high performance of SolaRes in *GHI* with the 1-minute resolution over at least a
year, making SolaRes consistent with scientific and industrial applications. Ruiz-Arias *et al.* [2013]
also show high spatial variability, with *MBD* reaching -1% on two sites. Similarly, **Sect. 5.1** also
presents 0.4% difference in *MBD* between Lille and Palaiseau.

Scores in *DNI$_{pyr}$* slightly improve with a *RMSD* of 2.0% and a *MBD* of -1.2% with the Garcia cloud-
screening method, showing that the simpler approach based on the *AOT* data set is appropriate to
get high precision in *DNI$_{pyr}$*. Indeed *MBD* of -0.5% could be reached at Palaiseau with the *AOT* data
set and the Garcia cloud-screening method. The *RMSD* in *DNI$_{pyr}$* with SolaRes is twice larger than
presented by Ruiz-Arias *et al.* [2013], but for larger mean *AOT*. Ruiz-Arias *et al.* [2013] present
*MBD* of 0%, but which would be expected smaller as no circumsolar contribution is computed.

The improvement is not significant in *DifHI$_{pyr}$*, but we agree with the tendency of over estimation
as shown by Ruiz-Arias *et al.* [2013]. Moreover Ruiz-Arias *et al.* [2013] show spatial variability
and our scores for *DifHI$_{pyr}$* are similar to what is presented for one site, but where mean *AOT* is
smaller than at Lille in 2018. As inverted AERONET aerosol model is expected to be the best
aerosol model, the source of discrepancy can have other reasons, as the surface albedo. According
to AERONET inversion products, the surface albedo at Lille at 440 and 675 nm are smaller than
what is used here. Reducing the surface albedo is expected to reduce *DifHI*, as well as the *MBD*.
But studying the sensitivity on surface albedo is beyond the scope of this paper.

Anyway, the excellent *MBD* scores in *GHI* (**Table 9**) shows very efficient compensation between
*DNI* under-estimation and *DifHI* over-estimation. With this data set, both annual averages of *GHI$_{obs}$*
and *DifHI$_{obs}$* are closer to the averages by Gschwind *et al.* [2019]. As is mentioned earlier, such an
average is affected by mean *SZA*, which is indeed 52±13°, smaller than with the *AOT* input data set
(**Table 2**).

Table 9. As **Table 4** but for *GHI*, *DNI$_{pyr}$* and *DifHI$_{pyr}$*, at Lille in 2018, but with AERONET inverted
aerosol model.

| Solar resource parameter | Cloud-screening method | Number of comparison pairs | Mean solar resource parameters (W/m$^2$) | Comparison scores | |
|---|---|---|---|---|---|
| | | | | MBD (%) | RMSD (%) |
| GHI | Garcia | 26 500 | 581±193 | 0.2 | 1.2 |
| | L&A | 14 200 | 544±184 | 0 | 0.8 |
| DNI$_{pyr}$ | Garcia | 26 500 | 779±105 | -1.2 | 2.0 |
| | L&A | 14 200 | 808+-83 | -1.4 | 1.8 |
| DifHI$_{pyr}$ | Garcia | 26 500 | 105±40 | 7.1 | 9.5 |
| | L&A | 14 200 | 82±16 | 8.2 | 10.4 |

### 6.3. Impact of the input data source: reanalysis global data set

AERONET provides best precision and accuracy on observed column aerosol optical properties, but
the data set is site-defined and does not cover the entire globe. To provide solar resource parameters
anywhere on the globe, it is necessary to use a global data set defined on a regular grid and on a



constant time step, such as provided by CAMS and Modern-Era Retrospective Analysis for Research and Applications, Version 2 (MERRA-2) [Gelaro *et al.*, 2017] programs. As the disadvantage of such data sets compared to AERONET is their larger uncertainty, it is important to evaluate their influence on the computed solar resource components.


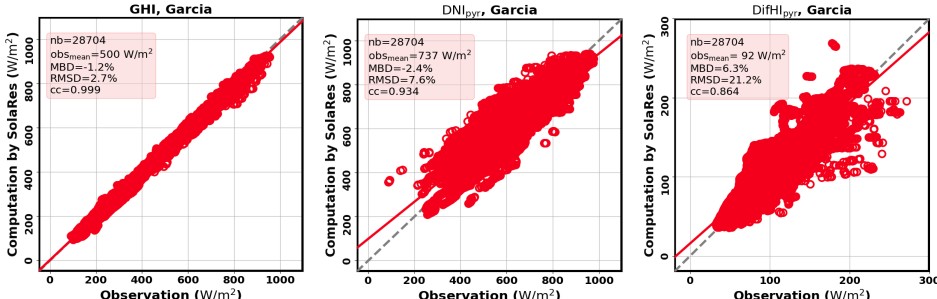

Figure 7. As *Fig. 2* for solar resource parameter comparisons at Lille but for CAMS-NRT as input data source instead of AERONET, with the Garcia cloud-screening procedure applied in 2018. $DNI_{pyr}$ and $DifHI_{pyr}$ are showed.


Comparison is performed at Lille in 2018 with CAMS-NRT (*Sect. 2.3*) instead of AERONET. The cloud-screening is now based uniquely on solar irradiance measurements, and not on AERONET. As CAMS-NRT is less precise than AERONET, and has less spatial and temporal resolution, the *RMSD* in the computed solar resource components increases, to reach 2.7% in *GHI* with the Garcia cloud-screening (*Fig. 7*). *RMSD* in *GHI* is 1.8% with the L&A cloud-screening (not shown), showing slightly more cloud-screening influence than 0.5% found with the AERONET *AOT* data set (*Sect. 5.1*). Consequently, the CAMS-NRT global data set increases the *RMSD* in *GHI* by 0.6 to 0.8%.

The impact is larger in $DNI_{pyr}$ and $DifHI_{pyr}$, with *RMSD* in $DNI_{pyr}$ increasing by ~5% to reach 7.6%, and *RMSD* in $DifHI_{pyr}$ increasing by more than 10%. This is consistent with Ruiz-Arias *et al.* [2013] stating that: *"the impact of aerosols in direct surface irradiance is about three to four times larger than it is in global surface irradiance"*, quoting Gueymard [2012]. Test was done by adding the Level 2.0 clear-sun cloud-screening, reducing *RMSD* in $DNI_{pyr}$ by only 0.3%. Witthuhn *et al.* [2021] shows that the increased *RMSD* for both *GHI* and *DNI* is caused by the dispersion of CAMS *AOT* compared to AERONET. Our results are similar to Witthuhn *et al.* [2021] in terms of *RMSD*, who give 8.6% *RMSD* in *DNI* but for all Germany in 2015, using CAMS reanalysis and a different cloud-screening procedure, 3.2% in *GHI* and 15.2% in *DifHI*. However their simulated *DNI* over estimates observation, even if their uncertainty source analysis suggests tendency for *DNI* underestimation, consistently with SolaRes results.

The *RMSD* in SolaRes *GHI* remains smaller than the best score of 3.0% provided by Sun *et al.* [2019]. The main differences with our comparison study, is that Sun *et al.* [2019] use the MERRA-2 data set instead of CAMS-NRT. Also, their scores are obtained for a much larger observation data set, more representative of the global variability of aerosol properties than the measurements of Lille and Palaiseau.






### 7. Conclusion

SolaRes is a tool to estimate solar resource components with precision and accuracy anywhere on the globe, in any meteorological and ground surface conditions, for any solar plant technology. SolaRes is designed for largest number of applications, from scientific to industrial, then producing time series at 1 minute time resolution and covering all situations for more than a year, with acceptable computing speed. SolaRes is based on radiative transfer computations with SMART-G,

and input parameters are atmospheric optical properties as the spectral aerosol and cloud optical thickness, which are usually available in many data sets. Computations are made on demand, in order to provide the best accuracy, and even interactions of the solar radiation field with 3D objects can be considered [Moulana *et al.*, Submitted].

The first step in the validation process consists in checking that SolaRes is able to reproduce the influence of aerosols and water vapour in clear-sky conditions. Indeed, aerosols and water vapour

are always present in the atmosphere, even in overcast conditions, and aerosols are the main factor of solar resource variability in clear-sky conditions, when *GHI* and *DNI* are maximum. Moreover aerosol and water vapour parameters can be measured precisely by ground-based instrumentation. AERONET provides such measurements, and the validation in clear-sky conditions is then a radiative closure study.

Comparisons between SolaRes estimates and ground-based observations are performed at 1 minute time resolution. Measurements are made in 2018-2019 by pyranometers and pyrheliometers mounted at Lille and Palaiseau both located in northern France. Measurements at Lille are made on the ATOLL platform and measurements at Palaiseau contribute to BSRN. $GHI_{obs}$ is slightly underestimated by SolaRes by 0.1% with a mean *RMSD* of around 1.0%, with a strict cloud-

screening method based on Long and Ackerman [2000] (L&A), but also filtering conditions with largest *AOT*, as those occurring in spring and summer. Another cloud-screening method based on Garcia *et al.* [2014] (Garcia thereafter) is used which is more representative of the aerosol variability conditions. Under estimation slightly worsens to 0.4% at Palaiseau and 0.8% at Lille, partly because of residual clouds increasing *DifHI*, and *RMSD* increases to ~1.6%, but for

conditions more representative of the mean aerosol conditions. Results are similar with another instrument operating in a slightly restricted spectrum.

SolaRes also performs well to reproduce the angular features of the solar radiation field. The comparison scores in both *DNI* and *DifHI* improve by considering the circumsolar contribution. Under-estimation of $DNI_{obs}$ by SolaRes decreases by 1% to reach a *MBD* of -1.0%, and the *RMSD*

slightly decreases to reach ~2%. Over-estimation of *DifHI* by SolaRes decreases by ~4% to reach a *MBD* of 3% at Lille and 2% at Palaiseau, with a *RMSD* of 10%. It is interesting to note that *DNI* under-estimation and *DifHI* over-estimation mostly compensate to provide mean agreement in *GHI*.

The advantages of using SolaRes for solar resource estimates with tilted panels is twofold: 1) *DNI* and *DifHI* are correctly computed, even considering the circumsolar contribution for comparison

purposes with observation; 2) *DifTI* can be computed by radiative transfer computations without using parameterisation of *DifHI*. Comparisons with observations made in a vertical plane facing South show satisfying agreement for *DifTI* with a *RMSD* of 8%. It is suggested a strong influence of reflection by not only ground surface but also surrounding buildings, and changing with the season. Indeed, *GTI* could be reproduced with same scores as *GHI* but with a surface albedo increased from

0.13 to 0.35. More studies are necessary for inferring the effective value of ground surface and building surface albedo.

An *AOT* data set allows to constrain the mean aerosol extinction as well as the mean aerosol size (by the spectral dependence of the aerosol extinction), but not the aerosol absorption neither the angular behaviour of aerosol scattering. Hypothesis is then necessary to complement the aerosol

model in order to perform radiative transfer computations. Two aerosol models of the OPAC





database are combined to reproduce spectral *AOT*. The aerosol models are modified to show the sensitivity of the solar resource parameters on these hypothesis. Spectral *AOT* efficiently constrains *DNI* which is little sensitive to the aerosol models, while *DifHI* is highly sensitive. SolaRes *DifHI* significantly decreases with increasing aerosol absorption of the fine aerosol model, and over-estimation changes to under-estimation with urban aerosols instead of continental clean aerosols. Consequently *GHI* under-estimation could worsen to 2% and *RMSD* in GHI could increase to 4%. The best combination at Lille and Palaiseau consists in a continental clean aerosol model mixed with a desert dust model. Tests with the aerosol models inverted by AERONET, then defining aerosol absorption and angular scattering, show significant improvement in scores in *GHI*, by decreasing *MBD* to 0.2% and by decreasing *RMSD* by 0.5%. *RMSD* in *GHI* could be smaller than 1% at Lille with the L&A cloud-screening.

Comparisons are also done in the SolaRes global mode, by using *AOT* and *WVC* delivered by CAMS-NRT instead of AERONET. The *RMSD* in *GHI* becomes 1.8% with the L&A cloud-screening and 2.7% with the Garcia cloud-screening, increasing by 0.6 to 1.0%. The *RMSD* in *DNI* increases by ~5%, and the *RMSD* in *DifHI* increases by more than 10%. The scores worsen as expected, because of modelling errors and rawer resolution in space and time, but with the strong advantage to cover the entire globe for many years, which is not possible with AERONET.

Scores also depend on the site. The combined irradiance and AERONET cloud-screening methods show that there are ~2% more clear-sky conditions at Palaiseau than at Lille, also smaller *AOT* by ~0.02 and smaller *AOT* variability, consequently slightly larger $DNI_{obs}$. Comparison scores are better at Palaiseau, by ~0.2% in *RMSD* in *GHI* and 0.4% in *MBD*.

Perspectives consist in validating SolaRes in more diverse conditions, as in arid environment strongly affected by desert dust, as already done for *DNI* with the ASoRA method [Elias *et al.*, 2021]. More studies are also necessary for computations in tilted planes, investigating on the influence of environment by reflection of the solar radiation. SolaRes may be improved by considering the spectral dependence of surface albedo, and even bidirectional reflectance distribution function. Furthermore, SolaRes in global mode will be tested in all-sky conditions.





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

**Acknowledgements:** LOA staff is acknowledged for providing the observation data set of Lille
composed by GHI, DNI, DifHI and GTI, as well as BSRN for the observation data set of Palaiseau made of GHI, DNI and DifHI. AERONET is also acknowledged for the data set of AOT, WVC and inverted aerosol model products, as well as Philippe Goloub as PI of the Lille and Palaiseau stations. And CAMS is also acknowledged for providing the AOT data.

**Competing interests:** The contact author has declared that none of the authors has any competing
interests.

**Author contribution:** GC wrote the Section 4 and developed the cloud-screening codes. TE and MM developed the SolaRes code and TE made the SolaRes computations for the paper. TE wrote
the other Sections of the manuscript. NF is the Lille instrument PI. NF and IC are the Ph. D. supervisors of GC. All authors contributed on discussions about the work in progress, and all authors made comments on the paper writing.