# Peer review of "Regional validation of the solar irradiance tool SolaRes in clear-sky conditions, with a focus on the aerosol module"

_Atmospheric Measurement Techniques, 2023_

## Author Comment (AC1)

AMT-2023-236 | Research article
Submitted on 09 Nov 2023
Regional validation of the solar irradiance tool SolaRes in clear-sky conditions, with a focus on the aerosol module
Thierry Elias, Nicolas Ferlay, Gabriel Chesnoiu, Isabelle Chiapello, and Mustapha Moulana

General Comments:

This manuscript compares the solar irradiance under clear sky conditions simulated by the SolaRes (solar resource estimation tool) at two sites with ground-based solar irradiance observations, evaluating the impact of different cloud-screening procedures and the incorporation of aerosol optical characteristics from different sources on the simulation results. The study indicates that SolaRes performs well in the study area when using AERONET AOT. The research content aligns with the publication scope of the Atmospheric Measurement Techniques journal, and **publication is recommended after revisions**.

Thank you for your comments. Please find our answers below each of the comment (in bold).

**Specific Comments:**

**1) The English grammar and structure of the manuscript need further improvement. Excessive forward and backward references make the article difficult to read. The abstract contains a lot of content but does not clearly explain the purpose of the research.**

English grammar has been revised in this new version, as well as the abstract and many parts of the manuscript, as can be seen in the version with corrections.

We removed some "**forward and backward references**" for clarity. We also hpe that the changes improve the structure.

**2) When assessing the impact of aerosol optical properties on the simulated results in the manuscript, the aerosol scattering phase function and single scattering albedo (SSA) are derived from different aerosol models. Due to the influence of various factors such as aerosol size distribution, chemical composition, hygroscopicity, morphology, etc., and their values vary with altitude, a thorough analysis of the corresponding errors should be conducted.**

To resolve the radiative transfer, optical properties are required, as the aerosol scattering and absorbing properties. Some of the required aerosol optical properties can be delivered by measurements, as aerosol optical thickness by AERONET. However, measurements rarely provide all necessary optical properties at all wavelengths. Strategies are then required to extrapolate the measurements. One approach is to estimate the microphysical properties reproducing the measurements, which are then used to compute any required optical properties, as done here with the 2-model mixing approach.

As SolaRes is fed with aerosol optical properties and not microphysical properties, we judge that a thorough analysis of the errors caused by microphysical properties (which are the size distribution,

the refractive index (describing the chemical nature), the shape, … ) is out of the scope of the paper. Anyway, some sensitivity studies are performed based on optical properties.

A sensitivity study of DNI is presented in a proceeding paper by Elias et al. [2019], and the main factor in DNI is AOT and a secondary factor is the Angstrom exponent. Witthuhn et al. [2021] show that the main contributor to RMSE in both GHI and DNI is AOT.

In the paper we show the influence of AOT by showing scores in different seasons (Section 5), and by using the CAMS source instead of AERONET (Section 6.3). Moreover, in the response of reviewer #1, we also show the sensitivity of the comparison scores to the AOT range. With the Garcia cloud-screening method, MBD and RMSD in GHI are -0.6% and 1.6%, respectively, for AOT < 0.2, and increase to -1.3% and 2.2%, respectively, for AOT>0.2.

In the paper, we also show that exploiting the spectral AOT (giving the Angstrom exponent) to model aerosols in SolaRes allows to sufficiently constrain the aerosol size distribution, given the good comparison scores in DNI and GHI, and that the AERONET-inverted model does not significantly improve the performance in DNI (Section 6.2).

We also show the influence of the aerosol single scattering albedo (SSA) by changing the aerosol models (Section 6.1), and it is significant in DifHI. For example MBD in DifHI varies from 2,2% to 12.3% when the large-$\alpha$ model changes from continental clean to urban (small- $\alpha$ model being desert dust).

The vertical profile of aerosol extinction has no influence on DNI, as it depends on the accumulated atmospheric extinction along the path, and has little influence on DifHI.

**3) The two sites used by the authors for evaluation are relatively similar and both belong to regions with low aerosol loading, thus their representativeness is limited. If observational data from other sites with different aerosol concentrations and sources could be included, it would contribute to a better assessment of the application of SolaRes.**

The answer here is equal to the answer to a comment by reviewer #1:

Our region of study, northern France is significantly influenced by anthropogenic aerosol sources, and particulate pollution episodes (Chebaicheb et al., 2023), especially those producing nitrates (Drugé et al., 2019) with a diversity of local and regional origins (Potier et al., 2021).

Aerosol variability is large at Lille and Palaiseau, with standard deviation in aerosol optical thickness (AOT) and in the Angstrom exponent ($\alpha$) reaching 70% and 30%, respectively. The mean AOT level is moderate with an annual average of 0.14 at 500 nm (Table 3), close to the European average according to Gueymard and Yang [2020], based on AERONET, which is larger than the average in North America. According to the Köppen–Geiger climate classification, both sites are affected by a climate similar to western Germany [Witthuhn et al., 2021], and similar to England, Ireland, Belgium, Netherlands (Cfb). The annual averages at Lille and Palaiseau are also close to the Cfb average [Gueymard and Yang, 2020]. AOT is 0.16 in spring-summer at Lille, the 90[th] percentile over a year is 0.32. There are a number and a variety of recorded aerosol events (as volcanic plumes, Derimian et al., 2012, Boichu et al., 2016), including heavy regional pollution events. For example in March 2014 in Lille and Palaiseau (Dupont et al., 2016, Favez et al., 2021),

measured AOD reached values up to 0.9, such that this can be classified as a severe aerosol pollution event, and such kind of events are recurrently observed in spring over this part of Europe.

Moreover, these two sites are appropriate to test the cloud-screening techniques, as the cloud influence is strong and highly variable in the region.

We consequently judge that these sites are good candidates to validate SolaRes in variable clear-sky conditions. But we agree that these sites are not fully representative of the global variability in terms of aerosol properties, it is why we chose to specify "regional validation" in the title. This article can be considered as a first step of a larger and comprehensive validation process of SolaRes, focusing on typical aerosol conditions of northern Europe in clear-sky conditions. Among our perspectives, we will consider additional aerosol conditions (in type and load) over other continents and also provide all-sky conditions evaluation of the algorithm.

The DNI approach (inspired from ASoRA) was already validated at Ouarazate, in Morocco, nearer the desert dust sources [Elias et al., 2021] than both Lille and Palaiseau.

A paragraph is added in Section 2 to justify the choice of these 2 sites.

Chebaicheb, H., Joel F. de Brito, Gang Chen, Emmanuel Tison, Caroline Marchand, et al.. Investigation of four-year chemical composition and organic aerosol sources of submicron particles at the ATOLL site in northern France. Environmental Pollution, 2023, 330, pp.121805. 10.1016/j.envpol.2023.121805 . Hal-04146486.

Drugé, T., Nabat, P., Mallet, M., and Somot, S.: Model simulation of ammonium and nitrate aerosols distribution in the Euro-Mediterranean region and their radiative and climatic effects over 1979–2016, Atmos. Chem. Phys., 19, 3707–3731, https://doi.org/10.5194/acp-19-3707-2019, 2019.

Gueymard, C. A., D. Yang, Worldwide validation of CAMS and MERRA-2 reanalysis aerosol optical depth products using 15 years of AERONET observations, Atmospheric Environment, 225, 2020, 117216, https://doi.org/10.1016/j.atmosenv.2019.117216.

Potier, E., Waked, A., Bourin, A., Minvielle, F., Péré, J., Perdrix, E., Michoud, V., Riffault, V., Alleman, L., and Sauvage, S.: Characterizing the regional contribution to PM10 pollution over northern France using two complementary approaches: Chemistry transport and trajectory-based receptor models, Atmospheric Research, 223, 1–14, https://doi.org/https://doi.org/10.1016/j.atmosres.2019.03.002, 2019.

**4) Line 793: "Small/larger aerosol model" is not a common expression.**

We propose to name these 2 categories of aerosol model as large- $\alpha$ and small-$\alpha$, $\alpha$ standing for the Angstrom exponent, instead of "small" and "larger", respectively.

---

## Author Comment (AC2)

**AMT-2023-236 | Research article**
Submitted on 09 Nov 2023
Regional validation of the solar irradiance tool SolaRes in clear-sky conditions, with a focus on the aerosol module
Thierry Elias, Nicolas Ferlay, Gabriel Chesnoiu, Isabelle Chiapello, and Mustapha Moulana

The authors validate SolaRes (Solar Resource estimate) solar irradiance (including multiple components of solar irradiance) in clear sky conditions using AERONET spectral aerosol optical depths (AOD) as input to the SMART-G radiative transfer model, comparing to ground-based solar irradiance measurements. The authors use CAMS-NRT to show that SolaRes can be more broadly applied globally. The authors present statistics indicating good agreement between SolarRes and using AERONET spectral AOD as input. Because this region of the planet tends to have minimal aerosol loading, the statistics presented (even though they are normalized) are heavily skewed towards lower aerosol loading days. **I recommend publication after revision**.

Thank you for your comments. Please find our answers below each of the comment (in bold).

**Major feedback:**

**1. I think the authors need to spend some time copy-editing for grammar/English peculiarities.**

English grammar has been revised, as can be seen in the version with corrections.

**2. The paper appears to be simple validation (which is a perfect fit for AMT) of SolaRes results with AERONET AOD and trace gas as input, comparing against in-situ irradiance observations.**

**After reading the abstract, the methodology was not immediately clear, as the only statement regarding validation was "Measurements for validation are made at two sites in Northern France." What are you measuring and what are you using to make these measurements (instruments)?**

We agree that the abstract was not clear enough, and the abstract has been deeply revised, with a lot of efforts to clarify our objective and methodology. We present the measurements (global, direct, diffuse surface solar irradiance) and instruments (pyranometer, pyreheliometer) used in the paper.

**3. Considering that a major point of this research is to show that SolarRes can reproduce irradiance regardless of atmospheric loading, you really should constrain your errors by AERONET AOD or AOD/mu0. E.g., errors are w for 0.0<AOD<0.15, x for 0.15<AOD<0.5, y for 0.5<AOD<1.0, and z for AOD>1.0. Otherwise, you are mostly presenting evidence to suggest that SolarRes works well when mean aerosol loading is low, because it is in the mean. Figure 4 is a good example of a plot that shows how both aerosol loading (AOD) and aerosol**

**type (ANG) are affecting your results, but this needs to be done with the errors using AERONET (and CAMS-NRT if you plan on extrapolating these results globally).**

Thank you for your suggestion.

Please find Table R1 with MBD and RMSD values in GHI for several AOT ranges. Performances depend little on the AOT range for most of the data set, when AOT values are included between 0.05 and 0.20 (75% of the data set), even if the averaged AOT significantly changes from 0.07 to 1.7. Indeed MBD is -0.06% or -0.07% and RMSD varies between 1.4 and 1.8% for AOT between 0.05 and 0.20. There may be some sensitivity to AOT beyond this AOT range, but to be confirmed with more data. Indeed MBD and RMSD fall down to -0.4% and 1.0%, respectively, for AOT < 0.05, but for only 2400 comparison pairs (~4% of the data set), and oppositely MBD and RMSD increase to -1.0% and -2.1%, resp., for 0.20 < AOT < 0.30, for 7800 comparison pairs (12%), and to -1.8% and 2.5%, resp., for 3600 comparison pairs (~6%).

To summarise, with the Garcia cloud-screening method, MBD and RMSD in GHI are -0.6% and 1.6%, respectively, for AOT < 0.2 for 80% of the data set, and increase to -1.3% and 2.2%, respectively, for AOT>0.2.

Table R1. Influence of the AOT range (500 nm) on MBD and RMSD in GHI (Garcia cloud-screening, Lille, 2018-2019)

| AOT range | Mean AOT | nb | Mean GHI (W/m2) | MBD (%) | RMSD (%) |
|---|---|---|---|---|---|
| All (as in Table 4) | 0.13+-0.08 | 50 000 | 500+-228 | -0.8 | 1.7 |
| | | | | | |
| AOT < 0.05 | 0,04+-0,005 | 2400 | 411+-152 | -0.4 | 1,0 |
| 0,05 < AOT < 0,10 | 0.07+-0,01 | 15 800 | 495+-237 | -0.6 | 1,4 |
| 0,10 < Aot < 0,15 | 0.12+-0,01 | 13 900 | 502+-223 | -0.7 | 1,8 |
| 0,15 < Aot < 0,20 | 0.17+-0,01 | 8400 | 520+-234 | -0.6 | 1,6 |
| 0,20 < Aot < 0,30 | 0,24+-0,03 | 7800 | 486+-216 | -1.0 | 2.1 |
| AOT > 0,30 | 0,38+-0,06 | 3600 | 550+-229 | -1.8 | 2.5 |
| | | | | | |
| **AOT < 0.20** | **0,11+-0,04** | **40 400** | **497+-228** | **-0.6** | **1.6** |
| **AOT > 0.20** | **0,28+-0,06** | **11 400** | **507+-222** | **-1.3** | **2.2** |

As suggested, we plotted Figure R1 similarly to Figure 4, but for GHI difference in function of both Angstrom exponent and AOT, but it is difficult to interpret. Table R1 is more informative.

[Figure]

Figure R1. Relative difference in GHI in function of the Ansgtrom exponent and the aerosol optical thickness.

**4. Considering how much high AOD data is screened out by the L&A cloud screening (Fig 2, DifHI), maybe just remove this technique from the paper with an explanation that the screening removed too many high AOD days? This could significantly shorten this paper.**

Validation is performed in uncloudy conditions as large uncertainties affect the measurements of the cloud properties on a local scale. In clear-sky conditions, satisfying precision can be reached on the aerosol optical thickness, which is the main factor on both DNI and GHI in clear-sky conditions, and which is provided by AERONET.

A cloud-screening procedure is then applied, in order to identify clear-sky moments, based on measurements of solar irradiance. It is however difficult to get rid of all overcast conditions, especially in an environment as Lille and Palaiseau, where high variety of cloud types (cumulus, low stratiform clouds, cirrus, multi-layering, …) generate heterogeneous 3D cloud fields. Residual clouds may then affect the selected measurements, which then may deviate from computations of clear-sky conditions. To get best performances, there is a tendency to use strict cloud-screening methods rejecting most residual clouds. However, such cloud-screening methods may also reject large AOT conditions, which have a similar effect to clouds, i.e. decreasing DNI and increasing DifHI. Best performances may indeed be reached with strict cloud-screening methods, but at the expense of the representativity in terms of aerosols, i.e. in conditions less turbid than usually encountered.

Here, two cloud-screening methods are applied, showing contrasting results in terms of comparison scores. A strict method provides the best scores but with reduced mean AOT compared to what is measured by AERONET (0.10 instead of 0.14). Another method is applied, which is more representative, with a mean AOT closer to the AERONET average (0.13), and keeping twice more moments, up to 50 000 in 2 years. However residual clouds may affect the data set and indeed the performances are degraded. The impact of the cloud-screening in GHI is 0.3% in MBD, and 0.5% in RMSD, at both Lille and Palaiseau.

Lille and Palaiseau are good candidates to check the influence of the cloud-screening procedures, as they witness variable overcast conditions, with 3D variable cloudy fields made of cumulus, stratus and fogs, cirrus, etc…

We then prefer keeping both cloud-screening techniques as an important result of the paper is that the cloud-screening technique affects the shown performance scores of a solar resource model in clear-sky conditions.

**5. Although the authors do tell us that this is a regional validation (it is in the title), I can't help but think that this would be significantly more impactful if other sites were also used for validation.**

The answer here is equal to the answer to the 3[rd] comment by referee #2:

The region of study, northern France is significantly influenced by anthropogenic aerosol sources, and particulate pollution episodes (Chebaicheb et al., 2023), especially those producing nitrates (Drugé et al., 2019) with a diversity of local and regional origins (Potier et al., 2021).

Aerosol variability is large at Lille and Palaiseau, with standard deviation in aerosol optical thickness (AOT) and in the Angstrom exponent ($\alpha$) reaching 70% and 30%, respectively. The mean AOT level is moderate with an annual average of 0.14 at 500 nm (Table 3), close to the European average according to Gueymard and Yang [2020], based on AERONET, which is larger than the average in North America. According to the Köppen–Geiger climate classification, both sites are affected by a climate similar to western Germany [Witthuhn et al., 2021], and similar to England, Ireland, Belgium, Netherlands (Cfb). The annual averages at Lille and Palaiseau are also close to the Cfb average [Gueymard and Yang, 2020]. AOT is 0.16 in spring-summer at Lille, the 90[th] percentile over a year is 0.32. There are a number and a variety of recorded aerosol events (as volcanic plumes, Derimian et al., 2012, Boichu et al., 2016), including heavy regional pollution events. For example in March 2014 in Lille and Palaiseau (Dupont et al., 2016, Favez et al., 2021), measured AOD reached values up to 0.9, such that this can be classified as a severe aerosol pollution event, and such kind of events are recurrently observed in spring over this part of Europe.

Moreover, these two sites are appropriate to test the cloud-screening techniques, as the cloud influence is strong and highly variable in the region.

We consequently judge that these sites are good candidates to validate SolaRes in variable clear-sky conditions. But we agree that these sites are not fully representative of the global variability in terms of aerosol properties, it is why we chose to specify "regional validation" in the title. This article can be considered as a first step of a larger and comprehensive validation process of SolaRes, focusing on typical aerosol conditions of northern Europe in clear-sky conditions. Among our perspectives, we will consider additional aerosol conditions (in type and load) over other continents and also provide all-sky conditions evaluation of the algorithm.

The DNI approach (inspired from ASoRA) was already validated at Ouarazate, in Morocco, nearer the desert dust sources [Elias et al., 2021] than both Lille and Palaiseau.

A paragraph is added in Section 2 to justify the choice of these 2 sites.

Chebaicheb, H., Joel F. de Brito, Gang Chen, Emmanuel Tison, Caroline Marchand, et al.. Investigation of four-year chemical composition and organic aerosol sources of submicron particles at the ATOLL site in northern France. Environmental Pollution, 2023, 330, pp.121805. 10.1016/j.envpol.2023.121805 . Hal-04146486.

Drugé, T., Nabat, P., Mallet, M., and Somot, S.: Model simulation of ammonium and nitrate aerosols distribution in the Euro-Mediterranean region and their radiative and climatic effects over 1979–2016, Atmos. Chem. Phys., 19, 3707–3731, https://doi.org/10.5194/acp-19-3707-2019, 2019.

Gueymard, C. A., D. Yang, Worldwide validation of CAMS and MERRA-2 reanalysis aerosol optical depth products using 15 years of AERONET observations, Atmospheric Environment, 225, 2020, 117216, https://doi.org/10.1016/j.atmosenv.2019.117216.

Potier, E., Waked, A., Bourin, A., Minvielle, F., Péré, J., Perdrix, E., Michoud, V., Riffault, V., Alleman, L., and Sauvage, S.: Characterizing the regional contribution to PM10 pollution over northern France using two complementary approaches: Chemistry transport and trajectory-based receptor models, Atmospheric Research, 223, 1–14, https://doi.org/https://doi.org/10.1016/j.atmosres.2019.03.002, 2019.

**Minor feedback:**

**P3, L65: "fastened" implies adhering to another object. "SMART-G run time is hastened…" makes more sense here.**

Thank you, this is corrected.

**P4, L113: "In the same field, AERONET aims to evaluate the aerosol radiative forcing, partly counteracting the greenhouse warming". This statement doesn't make sense for multiple reasons. As far as I understand it, AERONET can be used to account for aerosol direct radiative forcing only at local scales, which has nothing to do with counteracting greenhouse gas warming. Additionally, as far as I know, AERONET is primarily used to validate different satellite aerosol remote sensing algorithms.**

It is rewritten as: "In the same field, AERONET contributes to the estimate of the global aerosol radiative forcing by validating the aerosol satellite remote sensing retrievals and also aerosol climate models, in the context of the global greenhouse warming."

**P5, L115: This sentence doesn't make sense unless the word "are" was added by mistake.**

This sentence is rewritten as: "This paper thus presents a radiative closure study. Indeed two categories of independent simultaneously co-located measurements can be related by a radiative transfer code [e.g. Michalsky *et al.*, 2006; Ruiz-Arias *et al.*, 2013]."

**P5, L117: This statement depends on viewpoint. I recommend you modify it such that the statement is unambiguously correct, e.g., "From a radiation perspective, one of the main impacts of aerosols is to attenuate…". This still doesn't address indirect effects, but this paper is about clear-sky anyways.**

The sentence is rewritten as: "From a radiation perspective, one of the main impacts of aerosols is to extinguish the direct component of the solar radiation incident at surface level."

**P6, L167:  I recommend explaining what you mean by cosine errors at low sun angles.  Do you mean plane-parallel radiative transfer errors?**

We add for clarity that the cosine error that we evoke is about the pyranometer, which is used to measure the solar irradiance. Two references are added in the manuscript and it is rewritten as: "Observed global horizontal irradiance ($GHI_{obs}$) at Lille is obtained as the sum of direct and diffuse components, which is the preferred method for the measurement of global irradiance [Flowers and Maxwell, 1986], avoiding most cosine response's error of the instrument at low sun angles [Michalsky and Harrison, 1995; Mol *et al.*, 2024], and affected by smaller uncertainties in $GHI_{obs}$ than with unshaded instruments [Michalsky *et al.*, 1999]. The summation is indeed chosen by BSRN [Ohmura *et al.*, 1998], and can be expressed as:".

**P6, 211:  You should provide a reference to aerosol temporal variability since you assert that it is not highly autocorrelated.  I think you will find that AOD tends to have fairly high autocorrelation.**

We are sorry for the potential confusion of this paragraph.  We did not mean that AOD temporal signal is not highly autocorrelated. Our point is that aerosol properties are temporally significantly variable, so that in order to simulate clear-sky solar irradiance at a high temporal resolution and with accuracy, a maximal exploitation of the available information about temporal variability should be considered. As a reference about aerosol variability, please see Cheng *et al.* [2021] that estimates amean variability of 0.015 unity per hour for the AOT at Lille, and 0.035 for the mean Angstrom exponent variability.

**P6, L217: "..with possible inconvenient on solar resource precision." I don't know what you are implying here.  Do you mean that the L1.5 AERONET inversion data precision is insufficient?  I think you will find that the drivers of error for AERONET inversions will be scattering geometry and optical loading.  Good scattering geometry will only be found when the sun is low in the sky (morning or evening).**

**P6, L220:  A 2013 citation of AERONET SSA uncertainty can not be using V3 AERONET data.  I believe things are quite a bit different now as compared to V2.  I'd also recommend only using AERONET SSA if AOD is >0.2 or 0.3, as SSA inversions can become rather unreliable at lower aerosol loading.**

We answer here to the two comments which deal with similar subjects.

The measurement geometry is indeed important for SSA uncertainty, and the new V3 hybrid mode scan allows to make "*SSA retrievals to SZAs less than 50° to as small as 25°*" [Sinyuk et al. 2020]. Errors in the AERONET inverted parameters indeed also depend on AOT (proportional to aerosol loading). A further quality screening is applied in Level 2.0 compared to Level 1.5, and the Level 2.0 AOT is assumed to be more precise than the Level 1.5 AOT, which can then affect the SSA retrieval.

It is rewritten as: "In this context, although Level 2.0 inversion data (quality checked) is expected to be more precise than Level 1.5 data, we choose to use the Level 1.5 inversion data as other authors [Ruiz-Arias et al., 2013; Cheng et al., 2021; Witthuhn et al., 2021], as Level 2.0 inverted data set is

too sparse and limits the statistical significance of our assessment. Indeed Ruiz-Arias *et al.* [2013] mention an increase in uncertainty of Level 1.5 (V2) aerosol single scattering albedo (*SSA)* compared to Level 2.0, to the 0.05–0.07 range, while Witthuhn et al. [2021] mention an uncertainty of 0.03 for Level 1.5, consistently with an uncertainty of ±0.03 on the V3 Level 2 by Sinyuk *et al.* [2020]. The option "hybrid scan" [Sinyuk *et al.,* 2020] is chosen."

We agree that to reach the best precision, we could add a criterion on AOT, but as we wish to provide as many 1-minute estimates as possible, we choose to rely on Level 1.5 AERONET-inverted aerosol models, and we prefer not adding a criterion on AOT.  This approach eventually enables to significantly improve comparison scores in GHI.

**P7, L256:  "… by interpolating the aerosol extinction properties at 1 minute" maybe should be "…by first interpolating the aerosol extinction properties to the same 1-minute cadence.".**

It is rewritten as: "On the one hand, *DNI* is computed at the time resolution of 1 minute by interpolating aerosol optical thickness at 1 minute. On the other hand, *DifHI* is computed at 15-minute resolution by radiative transfer computations with SMART-G, to limit the computational time, and is then interpolated linearly at the 1-minute resolution."

**P9, L340:  Couldn't this use of plane parallel geometry dominate your errors in real-use scenarios when the sun is low in the sky?**

Yes indeed the plane-parallel is an approximation decreasing the computing time but increasing the errors for large SZA.  The impact of this approximation on computation errors is reduced by screening out cases for SZA>80°, as is done by other authors.  That further data-screening has little impact in terms of solar resource as DNI and DifHI are generally small as such large values of SZA.

**P10, 369:  Which two OPAC models, how are these chosen?**

One model must be 'small-α', and the other model 'large-α'.  It is consequently rewritten as: "To span a large range of Ångström exponent (α) values, it is recommended that one model is characterised by a large value of α and another by a smaller value of α. We then refer to a small- α model and to a large- α model."

As illustrated in Section 6.1, the candidate models are not unique.

**P10, L381:  AERONET does not observe anything close to 280 nm or 4000 nm.  Wouldn't this significantly affect results if aerosol loading was elevated?  I assume that you are just using OPAC model parameters at other wavelengths (280→4000 nm), but this needs to be explicitly stated.**

The maximum of the available solar radiation in the atmosphere is in the visible bandwidth where AERONET also performs the measurements. The solar radiation flux decreases to 280 or 4000 nm, reducing the importance of the aerosol parameterisation.  The satisfying comparison scores show

that it is appropriate to extrapolate the spectral dependence from 440-870 nm beyond this spectral interval. This extrapolation may indeed have a stronger impact for larger aerosol loading.

The sentence is rewritten as: "The weights $w_{AM1}$ and $w_{AM2}$ are obtained from **Eq. (13a) and (13b)**, and are used to compute the aerosol transmittance at other wavelengths of the 280-4000 nm spectral interval."

**P10, L384: I think you mean "The vertical profile of AOT decreases exponentially with a scale height of 2 km".**

Yes, it is changed accordingly.

**P11-P12: Your page numbering here is not working.**

Yes, indeed. This is strange. Sorry for that.

**P12: The Canary Islands are on the northern edge of the Saharan dust transport pathway, your northern France sites are not. They will likely see much higher aerosol loading than what you will.**

The cloud-screening method, inspired by the work of Garcia et al. [2014], was indeed modified to adapt the algorithm to the specific conditions of Northern France. Precisions were added to the text.

**P14, L467: Another reason to constrain differences by AERONET AOD.**

In Fig. R2, mean DifHI is plotted versus mean AOT in the AOT ranges given in Table R1. That confirms that mean DiFHI is dependent on mean AOT.

Mean DifHI and mean AOT of L&A are reproduced by the Garcia data set with AOT < 0.15.

Anyway, some residual clouds may also affect DifHI in the different AOT ranges.

[Figure]

Figure R2. Mean DifHI versus mean AOT in different AOT ranges, for the Garcia data set, as given in Table R1: AOT < 0.05, 0.05 < AOT < 0.10, 0.10 < AOT < 0.15, 0.15 < AOT < 0.20, 0.20 < AOT

< 0.30, AOT > 0.30.  The point is also plotted for AOT<0.15, to represent the conditions of the L&A data set.

**P14-15:  Table 3 and  Figure 1 all seem to indicate that the L&A screening is removing significantly more outliers than the Garcia method, and crucially AERONET.  Angstrom exponent standard deviation is probably not useful unless AOD is greater than 0.15 (due to propagation of errors), but your cloud screening is clearly removing at least some elevated aerosol cases.  I think if you overlay the histograms of AOD (for the 3 different AOD filters in Table 3), you may see this as well.  I'd recommend just removing the L&A analysis.**

Fig. R3 shows the occurrence frequency of AOT for the three data sets: no irradiance cloud-screening, Garcia cloud-screening, and L&A cloud-screening.  L&A cloud-screening rejects some of the cases with AOT > 0.2, and all cases with AOT larger than 0.6.  The Garcia cloud-screening rejects a few situations with AOT included between 0.2 and 0.5, and all cases with AOT > 0.8, which are rare according to AERONET.

[Figure]

Figure R3. Occurrence Frequency of AOT for the 3 data sets: all AERONET, with the Garcia cloud-screening, and with the L&A cloud-screening.

According to the answer to the comment #3, rejecting cases with AOT>0.2 may improve the agreement.

All this discussion seems to us important, and we then prefer keeping the 2 techniques to show their contrasted influence.

**P16, L536:  If you are summing from i=1 to N, I would think you should divide by N, not nb.**

Thank you, you are right. It is changed accordingly.

**P22, L716: I don't know how this statement is supposed to read, but it is not correct as written.**

It is rewritten as: "1. Most of the day around noon, the sun, positioned in the southern half-sky, faces the instrument, and is thus included in the instrument field of view. Both diffuse and direct radiation are then observed.

2. At both beginning and end of the day, the sun could be positioned behind the instrument in the northern half-sky, the instrument sensor then being in shadows. Only diffuse radiation is observed, which is less dependent on SZA than direct radiation, generating the flatter wings at the end of the day than around noon while in the first regime, both diffuse and direct radiation contribute to the observed signal. .

Comparisons are made in both regimes independently."

**P23, L740: What surface albedo from MODIS? Integrated, SW, LW + SW?**

Surface albedo is taken from the CAMS-radiation service for the Lille and Palaiseau sites, which is inferred from MODIS satellite observation, as described in Section 2.3. It is broadband surface albedo in the solar spectral range (SW).

The last paragraph in Section 2.4 is written as: "CAMS-NRT data time series at Lille and Palaiseau are also downloaded from the CAMS-radiation service[footnote 3]. The 'research mode' allows to download not only *GHI*, *DNI*, and *DifHI*, but also the input data for the model, such as the solar broadband surface albedo, which is derived from the Moderate Resolution Imaging Spectroradiometer (MODIS) as described by Lefèvre *et al.* [2013]. It is a combination of the white-sky and black-sky albedos, in function of the proportion of the direct radiation in the global radiation [Lefèvre *et al.*, 2013]. Daily averages are computed, varying between 0.12 in November-December and 0.16 in June-July at Lille and Palaiseau, and are used as input in SolaRes radiative transfer simulations. Constant value is used by Lindsay et al. [2020], which is slightly larger than values used here for Palaiseau: "*broadband surface albedo [...] set to 0.2, a typical broadband value for grassland*"."

**P24, L751: I thought DNI was computed by using ancillary input AOD and modeling the incident radiation field, or through observation?**

Yes its computed using input AOT. The introduction of Section 6 is rewritten and this mention is removed.

**P24, L754: I would change this to "These parameters can not be provided by observations alone, and the direct-sun measurements only partially describe the necessary input aerosol properties."**

Thank you, it is changed accordingly.

**P24, L750: You need a citation or evidence to suggest that there is a high time variability of aerosol properties.**

L 760

The beginning of Section 6 is rewritten, and the mention of 'high time variability' is here erased.

**P24, L759: This should read "..with a time resolution coarser than…"**

Yes thank you. It is changed accordingly.

**P24 L788: You should identify the two OPAC models used earlier, most readers are not going to want to search down the text to find it.**

It is also mentioned at beginning of Section 5: "The continental clean and desert dust OPAC models are mixed to reproduce AERONET spectral *AOT*"

**P25, L821: AERONET provides information on aerosol size distribution (wavelength independent) and aerosol real/complex refractive indices. How are you extrapolating this information to (280→4000 nm)?**

Aerosol single scattering albedo is linearly interpolated between 440 and 1020 nm, and remains constant below 440 nm and above 1020 nm. The phase function at closest wavelength is used.

In Section 3, it is written as: "For the sensitivity study of *Sect. 6.2*, the AERONET inverted aerosol model provides the aerosol phase function and single scattering albedo at the four wavelengths of 440, 675, 870 and 1020 nm [Sinyuk et al., 2020]. In this case, *AOT* and the aerosol single scattering albedo (SSA) are linearly interpolated between 440 and 1020 nm, AOT is linearly extrapolated below 440 nm and above 1020 nm while SSA remains constant, and the phase function at the closest wavelength is used."

**P25, L825: I would change this to "…, with possible negative implications for solar resource precision."**

Thank you, it is changed accordingly.

**P27: Figure 7: I would plot the errors (model-observation) as a function of AOD, using different the z axis (color) to constrain by Angstrom exponent.**

Thank you for the suggestion.

Such a curve is plotted in Fig. R4, as we understand it. But it seems to us difficult to interpret. We choose not to add such a figure in the paper.

[Figure]

Figure R4. Difference in GHI plotted in function of AOT, with the Angstrom exponent in 3rd dimension.

**P28-29: I think you could shorten your conclusion section.**

We also made changes to the conclusion section, but We're afraid that it is not significantly shortened.

**P28, L915: I think Figure 4 is the only direct evidence you have shown that aerosol variability is important here. The statistics will wash away your high aerosol events.**

Figure 4 indeed shows the variability in the Angstrom exponent in function of the slant path AOT. Table 3 also shows high standard deviation in AOT (70%) and significant standard deviation in α (30%). AOT can reach high values but rather rarely, as for example AOT is larger than 0.3 in around 6% of the 1-minute moments at Lille in 2018-2019 (Table R1).

Figure 1 also shows the seasonal cycle of AOT, varying from less than 0.10 in winter to more than 0.15 in spring-summer.

**P28, L948: You may want to make the following change: '…,but not the aerosol absorption nor the angular…'**

Thank you. It was changed as: "but neither the aerosol absorption nor the angular behaviour of aerosol scattering"